# Modeling the Impact of Timeline Algorithms on Opinion Dynamics Using Low-rank Updates

## ABSTRACT

Timeline algorithms are key parts of online social networks, but during recent years they have been blamed for increasing the polarization and disagreement in popular social networks. One of the key obstacles to explaining these phenomena is that polarization and disagreement appear in a *global network-level*, whereas timeline algorithms operate on a *local user-level*. Bridging between these two levels of abstraction is a major challenge. In particular, while network-level polarization and disagreement have been successfully studied using opinion-formation models, it has remained an open question of how these models can be augmented to take into account the fine-grained impact of user-level timeline algorithms.

We make progress on this question by providing a way to model the impact of timeline algorithms on opinion dynamics. Specifically, we show how the popular Friedkin–Johnsen opinion-formation model can be augmented based on *aggregate information*, extracted from timeline data. Our idea is to combine the underlying follow-graph of the online social network with a graph that is induced by data from a timeline algorithm. The aggregate information that we consider are the topics that are discussed in the social network, as well as the users' interests and influence on these topics. To the best of our knowledge, this is the first work that allows to obtain theoretical guarantees for combining an opinion-formation model with a graph induced by a timeline algorithm.

We use our model to study the problem of minimizing the polarization and disagreement; we assume that we are allowed to make small changes to the users' timeline compositions by strengthening some topics of discussion and penalizing some others. We present a gradient descent-based algorithm for this problem, and show that under realistic parameter settings, our algorithm computes a $(1 + \epsilon)$-approximate solution in time $\widetilde{O}(m\sqrt{n}\log(1/\epsilon))$, where $m$ is the number of edges in the graph and $n$ is the number of vertices. We also present an algorithm that provably computes an $\epsilon$-approximation of our model in near-linear time. We evaluate our method on real-world data and show that it effectively reduces the polarization and disagreement in the network. We also show that our algorithm is orders of magnitude faster than a non-optimized black-box optimization approach. Finally, we release an anonymized graph dataset with ground-truth opinions and more than 27 000 nodes (the previously largest publicly available dataset contains less than 550 nodes).

*WWW'24, May 13–17, 2024, Singapore*
© 2023 Association for Computing Machinery.
ACM ISBN 978-1-4503-XXXX-X/18/06...$15.00
https://doi.org/10.1145/nnnnnnn.nnnnnnn

## CCS CONCEPTS

• **Information systems** → **Social networks**; • **Theory of computation** → **Graph algorithms analysis**; **Approximation algorithms analysis**.

## KEYWORDS

Opinion dynamics, social-network analysis, polarization, disagreement

**ACM Reference Format:**
. 2023. Modeling the Impact of Timeline Algorithms on Opinion Dynamics Using Low-rank Updates. In *Proceedings of Make sure to enter the correct conference title from your rights confirmation email (WWW'24)*. ACM, New York, NY, USA, 28 pages. https://doi.org/10.1145/nnnnnnn.nnnnnnn

## 1 INTRODUCTION

Online social networks are used by millions of people on a daily basis and they are integral parts of modern societies. However, during the last decade there has been growing criticism that timeline algorithms, employed in online social networks, create filter bubbles and increase the polarization and disagreement in societies.

Despite significant research effort, our understanding of these phenomena is still insufficient. One of the main challenges here is that polarization and disagreement appear at a *global network-level*, whereas timeline algorithms operate on a *local user-level*. So, on the one hand, opinion dynamics are commonly studied in the context of the graph structure of the social network. On the other hand, timeline algorithms provide a personalized ranking of content (such as posts on Facebook or Twitter) and only consider users' local neighborhoods in the graph (e.g., $k$-hop neighborhoods), without considering the global polarization and disagreement. Providing models that bridge the gap between these two levels of abstraction is a major challenge to facilitate our understanding of the underlying phenomena.

A popular way to study the network-level polarization and disagreement is using opinion-formation models, and one of the most popular abstractions is the Friedkin–Johnsen (FJ) model [14]. The vanilla version of the FJ model, however, is not sufficient to model real-world online social networks, since it assumes that the underlying graph is static, based only on friendship relations, and not taking into account *additional* relations and interactions based on recommendations from timeline algorithms.

To address these limitations a lot of attention has been devoted to augmenting the FJ model to understand phenomena that are more closely aligned with the real world [10, 11, 30, 34, 36, 39]. However, existing augmentations are rather simplistic: they either study a small number of edge additions or deletions [34, 39] or they directly perform global changes to the graph structure to minimize the polarization and disagreement [10, 11, 30] (see Section 2 for a more detailed description of existing approaches). Most importantly, these papers assume that the graph structure is manipulated

directly, which does not align with how timeline algorithms interact with the underlying graph structure. Hence, the augmentations studied in existing papers provide no way of incorporating the properties of timeline algorithms into opinion-formation models. They also provide no means of updating a timeline algorithm's recommendations to reduce polarization and disagreement.

**Our contributions.** In this paper, we make progress on these issues by introducing an augmentation of the FJ model that *combines a fixed underlying graph and a network that is based on aggregate information of a timeline algorithm.*

In particular, we obtain our *aggregate information* by aggregating along the topics that are discussed in the social network. First, for each user we consider how many posts of its timeline are from each topic. This provides us with the topic distribution on the timeline of each user. Second, for each topic we consider how frequently posts by the users are displayed by the timeline algorithm. This provides us with a distribution for each topic, indicating how influential each user is for this topic. We argue that this is a realistic way to obtain aggregate information for a large range of timeline algorithms in real-world platforms, e.g., on Twitter/$\mathbb{X}$ or Reddit.

Based on the aggregate information, we introduce a low-rank graph update, which encodes the social-network connections created by the timeline algorithm's recommendations. In other words, we use the aggregate information and the low-rank graph to bridge between the network-level opinion dynamics and the user-level recommendations of a timeline algorithm. Our model is the first that allows to quantify how timeline algorithms impact polarization and disagreement; we also show that our model can be computed in nearly-linear time. Details are presented in Section 4.

Next, we use our model to study how a timeline algorithm's recommendations need to be adapted to reduce polarization and disagreement, by allowing small changes to the aggregate information. More concretely, we allow small changes to the timelines of the users, such as reducing a user's interest in a highly polarizing topic and slightly strengthening a less controversial topic in the user's timeline. We believe that incorporating these types of the changes into real-world timeline algorithm is practical.

For this problem, we provide a gradient descent-based algorithm, called GDPM, and show that under realistic parameter settings it computes a $(1 + \epsilon)$-approximate solution in time $\widetilde{O}(m\sqrt{n} \log(1/\epsilon))$, where $n$ is the number of vertices and $m$ is the number of edges in the original graph. The details are presented in Section 5.2.

To obtain our efficient optimization algorithm, we have to overcome significant computational challenges. In particular, since it is possible that the number of edges introduced by the low-rank graph is much larger than in the original graph, even writing down the edges introduced by the recommender system may be infeasible in practice. Therefore, in Section 5.1 we show that we can efficiently approximate the opinions, the polarization, and the disagreement in time that is *near-linear* in the size of the *original* graph.

Furthermore, we experimentally evaluate our algorithm on 27 real-world datasets. Our results show that GDPM can efficiently reduce the *disagreement–polarization index* proposed by Musco et al. [30]. We also qualitatively evaluate which topics are favored and which topics are penalized when reducing the polarization and the disagreement. Additionally, our experiments show that our algorithms

are orders of magnitude faster than baseline algorithms and that they scale to graphs with millions of nodes and edges.

Finally, we make our code and two anonymized Twitter datasets available for research purposes in an anonymized repository [1]. Our anonymized graph datasets contain ground-truth opinions and the graph structure for more than 27 000 nodes. The previousy largest publicly available dataset contains less than 550 nodes [13].

We include all omitted proofs and our implementation in the appendix.

## 2 RELATED WORK

Over the past few years, researchers have studied the phenomena of political polarization on social media [20, 33]. The work includes understanding the impact of polarized discussions [3, 26] as well as developing mitigation strategies [2, 28].

From a practical point of view, there have been various attempts to develop algorithmic solutions to reduce polarization. Several works propose approaches that expose users to opposing viewpoints on online social networks [16, 17, 19]. Munson and Resnick [29] design a browser extension that visualizes the bias of a user's content consumption.

To study polarization in online social networks theoretically, researchers resorted to opinion formation models and in recent years the most popular model in this context is the the Friedkin–Johnsen (FJ) model [14]. It has been popular to augment the FJ model with abstractions of algorithmic interventions [5, 10, 38, 39]. Other works in this research area study the impact of adversaries [9, 15, 37] and viral content [36], as well basic properties of the FJ model [6].

Several works in this area dealt with the question of minimizing the polarization and disagreement using small updates to the underlying graph [27, 30, 39]. Zhu et al. [39] and Rácz and Rigobon [34] allow $k$ edge updates to the underlying graph. Musco et al. [30] allow to redistribute all edge weights arbitrarily, whereas Cinus et al. [11] allow edge updates under the constraints that the vertex degrees must stay the same and that no new edges are added to the graph. The main limitation of these works is that the graph updates performed in their algorithms have no clear correspondence with operations of timeline algorithms; for instance, it is unclear how the graph updates proposed in [11, 30] should be incorporated into a timeline algorithm. In contrast, we believe that incorporating the changes to the aggregate information that we study in this paper as feasible in practice. We believe that this is a significant contribution to this line of work.

## 3 PRELIMINARIES

**Linear algebra.** Let $G = (V, E, w)$ be an undirected, connected, weighted graph with $n = |V|$ vertices and $m = |E|$ edges. We set $\mathbf{L} = \mathbf{D} - \mathbf{A}$ to the Laplacian of $G$, where $\mathbf{D}$ is the diagonal matrix with $\mathbf{D}_{ii} = \sum_{j: (i,j) \in E} w_{ij}$ and $\mathbf{A}$ is the weighted adjacency matrix with $\mathbf{A}_{ij} = w_{ij}$.

For $\mathbf{X} \in \mathbb{R}^{n \times k}$, we denote the Frobenius norm by $\|\mathbf{X}\|_F = (\sum_{i,j} \mathbf{X}_{ij}^2)^{1/2}$. The spectral norm of $\mathbf{X}$ is $\|\mathbf{X}\|_2 = \sigma_{\max}(A)$, where $\sigma_{\max}(\mathbf{A})$ is the largest singular value of $\mathbf{A}$. We also use the 1-norm of a matrix, which is given by $\|\mathbf{X}\|_{1,1} = \sum_{ij} |\mathbf{X}_{ij}|$. We write $\mathbf{X}_i$ to denote the $i$-th row of $\mathbf{X}$.

We write $\mathbf{I}$ to denote the identity matrix and $\mathbf{1}$ to denote the vector with all entries equal to 1; the dimension will typically be clear from the context. Given a vector $\mathbf{v}$, we write $\mathrm{diag}(\mathbf{v})$ to denote the diagonal matrix with $\mathrm{diag}(\mathbf{v})_{ii} = \mathbf{v}_i$. For vectors $\mathbf{u}, \mathbf{v} \in \mathbb{R}^n$, we write $\mathbf{u} \odot \mathbf{v} \in \mathbb{R}^n$ to denote their Hadamard product, i.e., $(\mathbf{u} \odot \mathbf{v})_i = \mathbf{u}_i \mathbf{v}_i$. We define $\|\mathbf{v}\|_2 = (\sum_i \mathbf{v}_i^2)^{1/2}$ to be the Euclidean norm of $\mathbf{v}$. For a vector $\mathbf{v}$ and a convex set $Q$, $\mathrm{Proj}_Q(\mathbf{v})$ denotes the orthogonal projection of $\mathbf{v}$ onto $Q$.

We write $\mathrm{sgn}(x)\colon \mathbb{R} \to \{-, 0, +\}$ to denote the sign of $x$. We use the notation $\widetilde{O}(T)$ to denote running times of the form $T \log^{O(1)}(n)$. We write $\mathrm{poly}(n)$ to denote numbers bounded by $n^{O(1)}$.

**Friedkin–Johnsen (FJ) model.** Let $G$ and $\mathbf{L}$ be as defined above. In the FJ model [14], each node $i$ has a fixed *innate opinion* $\mathbf{s}_i$ and an *expressed opinion* $\mathbf{z}_i^{(t)}$ at time $t$. Initially, $\mathbf{z}_i^{(0)} = \mathbf{s}_i$, and at time $t + 1$ every node $i$ updates their expressed opinion as the weighted average of its own innate opinion and the expressed opinions of its neighbors:

$$\mathbf{z}_i^{(t+1)} = \frac{\mathbf{s}_i + \sum_{j\,:\,(i,j)\in E} w_{ij} \mathbf{z}_i^{(t)}}{1 + \sum_{j\,:\,(i,j)\in E} w_{ij}}. \tag{1}$$

We write $\mathbf{s} \in \mathbb{R}^n$ and $\mathbf{z}^{(t)} \in \mathbb{R}^n$ to denote the vectors of innate and expressed opinions, respectively. It is known that in the limit, the expressed opinions converge to $\mathbf{z} = \lim_{t\to\infty} \mathbf{z}^{(t)} = (\mathbf{I} + \mathbf{L})^{-1}\mathbf{s}$.

We assume that the innate opinions are mean-centered and in the interval $[-1, 1]$, i.e., $\sum_{i\in V} \mathbf{s}_i = 0$ and $\mathbf{s}_i \in [-1, 1]$ for all $i \in V$. The latter implies that $\mathbf{z}_i^{(t)} \in [-1, 1]$. We note that these assumptions are made without loss of generality as they can always be achieved by rescaling the opinions $\mathbf{s}$.

For mean-centered opinions, the *polarization index* $P(G)$ measures the variance of the opinions and is given by $P(G) = \sum_{i\in V} \mathbf{z}_i^2$. The *disagreement index* $D(G)$ describes the tension along edges in the network and is given by $D(G) = \sum_{(i,j)\in E} w_{ij}(\mathbf{z}_i - \mathbf{z}_j)^2$. Finally, the *disagreement–polarization index* $I(G)$, on which we will focus for the rest of the paper, is given by

$$I(G) = P(G) + D(G) = \mathbf{s}^\top (\mathbf{I} + \mathbf{L})^{-1}\mathbf{s}, \tag{2}$$

where the last equality was shown by Musco et al. [30]. They also observe that the function $f(\mathbf{L}) = \mathbf{s}^\top (\mathbf{I} + \mathbf{L})^{-1}\mathbf{s}$ is convex if $\mathbf{L} \in \mathcal{L}$ is from a convex set of Laplacians $\mathcal{L}$ [32].

## 4 PROBLEM FORMULATION

In this section, we formally introduce our augmented version of the FJ model. In particular, we show how we use a timeline algorithm's aggregate information to obtain a low-rank graph update for the FJ model. At a high level, we start with the initial adjacency matrix $\mathbf{A}$, which only contains interaction-information (such as who follows whom) and add an adjacency matrix $\mathbf{A_X}$ based on the aggregate information. We also state the optimization problem we study for minimizing the disagreement–polarization index.

The *aggregate information* that we consider is as follows. We consider $k$ different topics and two row-stochastic matrices $\mathbf{X} \in [0, 1]^{n\times k}$ and $\mathbf{Y} \in [0, 1]^{k\times n}$, i.e., $\sum_{j=1}^k \mathbf{X}_{ij} = 1$, for all $i = 1, \dots, n$, and $\sum_{r=1}^n \mathbf{Y}_{jr} = 1$, for all $j = 1, \dots, k$, and we assume $k \leq n$. Here, $\mathbf{X}$ models how user timelines are formed based on various topics; more concretely, we assume that $\mathbf{X}_{ij}$ is the fraction of posts in

**Table 1: Summary of our notation**

| Variable | Meaning |
|---|---|
| $G = (V, E)$ | Original graph, vertex set, edge set |
| $n$ | Number of vertices in the original graph |
| $m$ | Number of edges in the original graph |
| $k$ | Number of topics |
| $\mathbf{X}$ | User–topic matrix (variable of our algorithm) |
| $\mathbf{Y}$ | Influence–topic matrix (fixed) |
| $\mathbf{A}$ | Adjacency matrix of the original graph |
| $\mathbf{A_X}$ | Low-rank adjacency matrix based on aggregate information |
| $\mathbf{L}$ | Laplacian of the original graph |
| $\mathbf{L_X}$ | Laplacian of the low-rank graph |
| $\mathbf{s}$ | Innate opinions |
| $\mathbf{z}$ | Expressed opinions for the original graph |
| $\mathbf{z_X}$ | Expressed opinions after adding the low-rank update to the original graph |
| $\widetilde{\mathbf{z}}_\mathbf{X}$ | Approximation of $\mathbf{z_X}$ |
| $f(\mathbf{X})$ | Objective function value for $\mathbf{X}$ |
| $\mathbf{X}^{(L)}$ | Entry-wise lower bound for $\mathbf{X}'$ in Problem 2 |
| $\mathbf{X}^{(U)}$ | Entry-wise upper bound for $\mathbf{X}'$ in Problem 2 |
| $\theta$ | Parameter used to define $\mathbf{X}^{(L)}$ and $\mathbf{X}^{(U)}$ |
| $Q$ | Feasible set of matrices $\mathbf{X}$ with $\mathbf{X}^{(L)} \leq \mathbf{X} \leq \mathbf{X}^{(U)}$ |
| $C$ | Percentage of extra edge weight added by low-rank update |

user $i$'s timeline from topic $j$. The matrix $\mathbf{Y}$ models which users are recommended by the timeline algorithm for each topic; that is, when the algorithm recommends contents for topic $j$, then a fraction of $\mathbf{Y}_{jr}$ of the contents was composed by user $r$.

Observe that if we consider the product $\mathbf{XY}$, a $(\mathbf{XY})_{ij}$-fraction of the recommended contents in the timeline of user $i$ is composed by user $j$. This can also be viewed as the impact that a user $j$ has on another user $i$. Since in general $\mathbf{XY}$ is a non-symmetric matrix, we also add the transposed term $\mathbf{Y}^\top \mathbf{X}^\top$, which ensures symmetry of the adjacency matrix. This can be interpreted as the impact of users' audience to them, for instance, users want to create content that is liked by their audience.

Thus, we will consider a scaled version of $\mathbf{XY} + \mathbf{Y}^T\mathbf{X}^T$. In the following lemma, we show that this matrix adds (weighted) edges of total weight $2n$.

**Lemma 1.** *It holds that* $\left\|\mathbf{XY} + \mathbf{Y}^T\mathbf{X}^T\right\|_{1,1} = 2n$.

To obtain a more fine-grained control over how many edges we add to the original graph, we consider a scaled version of $\mathbf{XY} + \mathbf{Y}^T\mathbf{X}^T$. More concretely, based on the result from Lemma 1, we add the low-rank adjacency matrix given by

$$\mathbf{A_X} = \frac{CW}{2n}\left(\mathbf{XY} + \mathbf{Y}^T\mathbf{X}^T\right),$$

where $C > 0$ is a parameter that is fixed throughout the paper and $W = \sum_{(i,j)\in E} w_{ij}$ is the total weight of edges in the original graph $G$. Observe that Lemma 1 implies that $\|\mathbf{A_X}\|_{1,1} = CW$ and thus if we add the edges in $\mathbf{A_X}$ to the graph, the total weight of edges increases by a $C$-fraction.[1] In practice, it may be realistic to think of $C = 10\%$ or $C = 50\%$.

---

[1]We note that while here we only guarantee that the *global* increase of edges is a $C$-fraction, in Figure 8 we show that also on a *local* user-level, the increase does not deviate a lot from 10%.

After adding the edges $\mathbf{A_X}$, which are based on the aggregate information, the new adjacency matrix becomes

$$\mathbf{A} + \mathbf{A_X} = \mathbf{A} + \frac{CW}{2n}\left(\mathbf{XY} + \mathbf{Y}^T\mathbf{X}^T\right),$$

where $\mathbf{A}$ is the adjacency matrix of the original graph and $\mathbf{A_X}$ is the adjacency matrix of the edges that are introduced by the low-rank update. Next, we write

$$\mathbf{L_X} = \mathrm{diag}(\mathbf{A_X}\mathbf{1}) - \mathbf{A_X}$$

to denote the Laplacian associated with the adjacency matrix $\mathbf{A_X}$. Note that the Laplacian of the combined graph is given by $\mathbf{L} + \mathbf{L_X}$, where $\mathbf{L}$ is the Laplacian of the original graph that only contains the follow-information.

Now, after adding the edges from the low-rank update, the expressed equilibrium opinions that are produced by the FJ opinion dynamics are given by $\mathbf{z_X} = (\mathbf{I} + \mathbf{L} + \mathbf{L_X})^{-1}\mathbf{s}$.

Next, we formally introduce the optimization problem that we study. Intuitively, the problem states that we wish to minimize the disagreement–polarization index (Eq. (2)), while allowing small changes to the aggregate information. In particular, we allow to make changes to how the users' timelines are composed of different topics. The formal definition is as follows.

**Problem 2.** *Given a graph $G = (V, E)$ with adjacency matrix $\mathbf{A}$ and Laplacian $\mathbf{L}$, user–topic matrix $\mathbf{X} \in [0, 1]^{k \times n}$, influence–topic matrix $\mathbf{Y} \in [0, 1]^{k \times n}$, and lower and upper bound matrices $\mathbf{X}^{(L)}$ and $\mathbf{X}^{(U)}$, respectively, find a matrix $\mathbf{X}' \in [0, 1]^{n \times k}$ to satisfy*

$$\min_{\mathbf{X}'} \quad f(\mathbf{X}') = \mathbf{s}^T (\mathbf{I} + \mathbf{L} + \mathbf{L_{X'}})^{-1} \mathbf{s},$$

$$\text{such that} \quad \left\|\mathbf{X}'_i\right\|_1 = 1, \quad \text{for all } i = 1, \dots, n, \text{ and} \quad (3)$$

$$\mathbf{X}^{(L)} \le \mathbf{X}' \le \mathbf{X}^{(U)}.$$

In Problem 2, we write $\mathbf{X}'_i$ to denote the $i$-th row of the matrix-valued variable $\mathbf{X}'$. The first constraint ensures that $\mathbf{X}'$ is a row-stochastic matrix. Furthermore, the matrices $\mathbf{X}^{(L)} \in [0, 1]^{n \times k}$ and $\mathbf{X}^{(U)} \in [0, 1]^{n \times k}$ are part of the input and they give entry-wise lower and upper bounds for the entries in $\mathbf{X}'$, i.e., we require $0 \le \mathbf{X}^{(L)}_{ij} \le \mathbf{X}'_{ij} \le \mathbf{X}^{(U)}_{ij} \le 1$ for all $i, j$. This constraint can be interpreted as a quantification of how much we can increase/decrease the attention of user $i$ to topic $j$ without the risk of making non-relevant recommendations and without violating ethical considerations. We further assume that $\mathbf{X}^{(L)} \le \mathbf{X} \le \mathbf{X}^{(U)}$, which corresponds to the assumption that the initial matrix $\mathbf{X}$ is a feasible solution to our optimization problem.

In the following, we let $Q$ denote the set of all matrices $\mathbf{X}'$ that satisfy the constraints of Problem 2. Observe that $Q$ is a convex set, since it is the intersection of a box and a hyperplane (the first constraint is equivalent to the hyperplane constraint $\langle \mathbf{X}'_i, \mathbf{1} \rangle = 1$, since all entries in $\mathbf{X}'$ are in the interval $[0, 1]$; the second constraint is a box constraint). Furthermore, observe that the constraints are independent across different rows $\mathbf{X}'_i$, which we will exploit later.

Since the objective function and $Q$ are convex, Problem 2 can be solved optimally in polynomial time. However, if we use a blackbox solver for this purpose, its running time will be prohibitively high in practice (see Section 6). Even more, already *a single* computation of the gradient is impractical when done naïvely (see Section 6). We address these challenges in the following section.

## 5 OPTIMIZATION ALGORITHM

In this section, we present a gradient-descent algorithm, which converges to an optimal solution for Problem 2. We present bounds for its running time and its approximation error after a given number of iterations. We also show that we can approximate the expressed opinions $\mathbf{z_X}$ highly efficiently. We conclude the section by presenting two greedy baseline algorithms.

### 5.1 Efficient estimation of expressed opinions

To understand the impact of the low-rank update on the user opinions, it is highly interesting to inspect the expressed opinions $\mathbf{z_X}$: comparing them with the original expressed opinions $\mathbf{z}$ will offer us insights into the impact of the timeline algorithm. However, even though in Lemma 1 we bound the total *weight* of edges that are added, their *number* could still be $\Omega(n^2)$, since the matrix $\mathbf{A_X}$ might be dense. Thus, even writing down $\mathbf{A_X}$ would result in running times of $\Omega(n^2)$ and would be prohibitively expensive. Therefore, one challenge is to show how to compute $\mathbf{z_X}$ efficiently.

In the following proposition, we show that since $\mathbf{A_X}$ has small rank, we can exploit the Woodbury identity to obtain an approximation $\widetilde{\mathbf{z}}_\mathbf{X}$ via Algorithm 1 (see Appendix 1 for the pseudocode). By using such an approximation we can achieve much faster running times, while still obtaining provably small errors. In the following proposition we use $\mathbf{U} = \begin{pmatrix} \mathbf{X} & \mathbf{Y}^\top \end{pmatrix}$ and $\mathbf{V} = \begin{pmatrix} \mathbf{Y} \\ \mathbf{X}^\top \end{pmatrix}$.

**Proposition 3.** *Let $\epsilon > 0$. Suppose $\left(-\frac{2n}{CW}\mathbf{I} + \mathbf{VM}^{-1}\mathbf{U}\right)^{-1}$ exists and $\left\|\mathbf{VM}^{-1}\mathbf{U}\right\|_2 \le 0.99\frac{2n}{CW}$. Algorithm 1 computes $\widetilde{\mathbf{z}}_\mathbf{X}$ with $\left\|\widetilde{\mathbf{z}}_\mathbf{X} - \mathbf{z_X}\right\|_2 \le \epsilon$ in expected time $\widetilde{O}((mk + nk^2 + k^3)\log(W/\epsilon))$.*

PROOF SKETCH. The algorithm is based on the observation that using the Woodbury matrix identity with $\mathbf{M} = \mathbf{I} + \mathbf{L} + \mathrm{diag}(\mathbf{A_X}\mathbf{1})$, and $\mathbf{U}$ and $\mathbf{V}$ as before, we get that

$$\mathbf{z_X} = \mathbf{M}^{-1}\mathbf{s} + \frac{CW}{2n}\mathbf{M}^{-1}\mathbf{U}\left(\mathbf{I} - \frac{CW}{2n}\mathbf{VM}^{-1}\mathbf{U}\right)^{-1}\mathbf{VM}^{-1}\mathbf{s}.$$

Now Algorithm 1 (pseudocode in the appendix) basically computes this quantity from right to left. Our main insight here is that we can compute the quantities $\mathbf{M}^{-1}\mathbf{s}$ and $\mathbf{M}^{-1}\mathbf{U}$ using the Laplacian solver from Lemma 10. Here, we approximate $\mathbf{M}^{-1}\mathbf{U}$ column-by-column using the call $\text{Solve}(\mathbf{M}, \mathbf{w}_j, \epsilon_\mathbf{R})$, where $\mathbf{w}_j$ is the $j$'th column of $\mathbf{U}$ and $\epsilon_\mathbf{R}$ is a suitable error parameter. The remaining matrix multiplications are efficient since $\mathbf{U}$ has only $2k$ columns and since matrix $\mathbf{V}$ has only $2k$ rows.

To obtain our guarantees for the approximation error, we have to perform an intricate error analysis to ensure that errors do not compound too much. This is a challenge since we solve $\mathbf{I} - \frac{CW}{2n}\mathbf{VM}^{-1}\mathbf{U}$ only approximately but then we have to compute an inverse of this approximate quantity. In the proposition we used the assumptions that $\mathbf{VM}^{-1}\mathbf{U}$ exists and that $\left\|\mathbf{VM}^{-1}\mathbf{U}\right\|_2 \le 0.99 \cdot \frac{2n}{CW}$, to ensure that this can be done without obtaining too much error. In the proof we will also show that these assumptions imply that the inverse $\mathbf{S}^{-1}$ used in the algorithm exists. See Appendix C.4 for details. □

The input of Algorithm 1 are the innate opinions $\mathbf{s}$, the user–topic matrix $\mathbf{X}$, the influence–topic matrix $\mathbf{Y}$, the fraction of weight parameter $C$, and the approximation error parameter $\epsilon$. The algorithm

returns the approximated expressed opinions $\widetilde{\mathbf{z}}_\mathbf{X}$. Note that if we consider the practical scenario of $k = \text{poly}\log(n)$ and $W \leq \text{poly}(n)$, the running time of Algorithm 1 is $\widetilde{O}(m\log(1/\epsilon))$.

Proposition 3 also allows us to efficiently evaluate the disagreement–polarization index after adding the edges in $\mathbf{A}_\mathbf{X}$. More concretely, in the following corollary we show that we can efficiently evaluate our objective function $f(\mathbf{X}) = \mathbf{s}^T (\mathbf{I} + \mathbf{L} + \mathbf{L}_\mathbf{X})^{-1} \mathbf{s}$ with small error.

**Corollary 4.** *Let $\epsilon > 0$. Suppose $\left(-\frac{2n}{CW}\mathbf{I} + \mathbf{V}\mathbf{M}^{-1}\mathbf{U}\right)^{-1}$ exists and $\left\|\mathbf{V}\mathbf{M}^{-1}\mathbf{U}\right\|_2 \leq 0.99\frac{2n}{CW}$. We can compute a value $\widetilde{f}$ such that $\left|\widetilde{f} - f(\mathbf{X})\right| \leq \epsilon$ in expected time $\widetilde{O}((mk + nk^2 + k^3)\log(W/\epsilon))$.*

## 5.2 Gradient descent-based polarization minimization

Next, we present our gradient descent-based polarization minimization (GDPM) algorithm. We start by presenting basic facts about the gradient of our problem in the following proposition.

**Proposition 5.** *The following three facts hold for the gradient of $f(\mathbf{X})$ with respect to $\mathbf{X}$:*
*(1) The gradient $\nabla_\mathbf{X} f(\mathbf{X})$ is given by*

$$\nabla_\mathbf{X} f(\mathbf{X}) = \frac{CW}{2n}\left(2 \cdot \mathbf{z}_\mathbf{X} \cdot \mathbf{z}_\mathbf{X}^\top \cdot \mathbf{Y}^\top - (\mathbf{z}_\mathbf{X} \odot \mathbf{z}_\mathbf{X}) \cdot \mathbf{1}_k^\top - \mathbf{1}_n \cdot (\mathbf{z}_\mathbf{X}^\top \odot \mathbf{z}_\mathbf{X}^\top) \cdot \mathbf{Y}^\top\right). \quad (4)$$

*(2) The function $f(\mathbf{X})$ is $L$-smooth with $L = \frac{8CW}{\sqrt{n}} \cdot \|\mathbf{s}\|_2 \cdot \|\mathbf{Y}\|_2^2$, i.e., for all $\mathbf{X}_1, \mathbf{X}_2 \in Q$ it holds that*

$$\|\nabla_\mathbf{X} f(\mathbf{X}_1) - \nabla_\mathbf{X} f(\mathbf{X}_2)\|_F \leq \frac{8CW}{\sqrt{n}} \cdot \|\mathbf{s}\|_2 \cdot \|\mathbf{Y}\|_2^2 \cdot \|\mathbf{X}_1 - \mathbf{X}_2\|_F.$$

*(3) Let $\epsilon > 0$. Suppose the conditions of Proposition 3 hold, then we can compute an approximate gradient $\widetilde{\nabla}_\mathbf{X} f(\mathbf{X})$ such that $\left\|\widetilde{\nabla}_\mathbf{X} f(\mathbf{X}) - \nabla_\mathbf{X} f(\mathbf{X})\right\|_F \leq \epsilon$ in expected time $\widetilde{O}((mk + nk^2 + k^3)\log(W/\epsilon))$.*

The gradient of our problem is given in Eq. (4) and in the second point we show that it is Lipschitz continuous. Computing the gradient exactly involves computing $\mathbf{z}_\mathbf{X}$ exactly; however, this requires to compute the matrix inverse $(\mathbf{I} + \mathbf{L})^{-1}$, which is expensive for large graphs. Hence, in the third point we show that an approximate gradient can be computed highly efficiently and with error guarantees.

Since we only have an approximate gradient, GDPM is an implementation of the gradient descent method by d'Aspremont [12], who analyzed a method of Nesterov [31] with approximate gradient. We use Kiwiel's algorithm [21] to compute the orthogonal projections $\text{Proj}_Q(\cdot)$ on our set of feasible solutions $Q$ in linear time, where we exploit that our constraints are independent across different rows of $\mathbf{X}$. The pseudocode of GDPM is given in Algorithm 2 in the appendix.

Algorithm 2 takes as input the innate opinions $\mathbf{s}$, the user–topic matrix $\mathbf{X}$, the influence–topic matrix $\mathbf{Y}$, the budget $\theta$, and the extra *weight* parameter $C$. It returns $\mathbf{X}^{(T)}$ after a number of iterations $T$.

In the following theorem we present error and running-time guarantees for GDPM, which show that it converges to the optimal solution given enough iterations.

**Theorem 6.** *Let $\epsilon > 0$. Suppose at each iteration of GDPM the conditions of Proposition 3 are satisfied. Then GDPM computes a solution $\mathbf{X}^{(T)}$ such that $f(\mathbf{X}) - f(\mathbf{X}^*) \leq \epsilon$ in expected time*

$$\widetilde{O}\left(\sqrt{\epsilon^{-1} \cdot CWkn} \cdot (mk + nk^2 + k^3)\log(W/\epsilon)\right),$$

*where $\mathbf{X}^*$ is the optimal solution for Problem 2.*

We note that in parameter settings that are realistic in practice, GDPM computes a solution with multiplicative error at most $(1+\epsilon')$ in time $\widetilde{O}(m\sqrt{n}\log(1/\epsilon'))$. More concretely, this is the case when the number of topics $k = \text{poly}\log(n)$ is small, the fraction of additional edges $C = O(1)$ is small, and the network is sparse with $W = \widetilde{O}(n)$. Additionally, it is realistic to assume that the optimal solution still has a large amount of polarization and disagreement since at least a constant fraction of the users will differ from the average opinion by at least 0.01; this argument implies that the polarization is at least $\mathsf{LB} = \Omega(n)$, which in turn implies that $f(\mathbf{X}^*) \geq \mathsf{LB} = \Omega(n)$. Hence, if in the theorem we set $\epsilon = \epsilon' \mathsf{LB}$, we get the bound above.

## 5.3 Baselines

Next, we introduce two greedy baseline algorithms. The baselines proceed in iterations and, intuitively, in each iteration they update the user timelines such that some topics are penalized and others are favored; the choice of these topics depends on the baseline.

More concretely, the baselines obtain as input the original graph and the matrices $\mathbf{X}, \mathbf{X}^{(L)}, \mathbf{X}^{(U)}, \mathbf{Y}$ and a number $T_{\max}$ of iterations to perform. First, we set $\mathbf{X}^{(0)} \leftarrow \mathbf{X}$. Now the algorithm performs $T_{\max}$ iterations. In each iteration $T$, we initialize $\mathbf{X}^{(T)} \leftarrow \mathbf{X}^{(T-1)}$. Then we manipulate the timeline of each user $i$ by redistributing the weights in row $i$ of $\mathbf{X}^{(T)}$. We pick two topics $j$ and $j'$ and transfer as much weight as possible from topic $j'$ to topic $j$. Intuitively, one can think of $j$ as a topic that we want to strengthen and $j'$ as a topic that we want to penalize; how these topics are picked depends on the implementation of the baseline (see below). To denote how much weight we can transfer, we set $\delta \leftarrow \min\{\mathbf{X}_{ij}^{(U)} - \mathbf{X}_{ij}^{(T)}, \mathbf{X}_{ij'}^{(T)} - \mathbf{X}_{ij'}^{(L)}\}$, i.e., $\delta$ corresponds to the weight that we can transfer from topic $j'$ to $j$ without violating the constraints of Problem 2. Then we set $\mathbf{X}_{ij}^{(T)} \leftarrow \mathbf{X}_{ij}^{(T)} + \delta$ and $\mathbf{X}_{ij'}^{(T)} \leftarrow \mathbf{X}_{ij'}^{(T)} - \delta$. As stated before, we do this for each user $i$. Then the next iteration $T + 1$ starts.

*Baseline 1: Strengthening non-controversial topics* (BL-1). We introduce our first baseline (BL-1), which aims to penalize controversial topics and to strengthen non-controversial topics. We build upon the meta-algorithm above and state how to pick the topics $j$ and $j'$ for the current user $i$. First, we compute $\widetilde{\mathbf{z}}_{\mathbf{X}^{(T)}}$ using Algorithm 1 and set $\bar{z} = \frac{1}{n}\sum_{u \in V} \widetilde{\mathbf{z}}_{\mathbf{X}^{(T)}}(u)$ to the average user opinion. Also, for each topic $j$ we set $\tau_j = \sum_{u \in V} \mathbf{Y}_{ju}\widetilde{\mathbf{z}}_{\mathbf{X}^{(T)}}(u)$ to the weighted average of the opinions of influential users for topic $j$. Since this does not depend on the user $i$, this can be done at the beginning of each iteration $T$. In BL-1, we set $j$ to a controversial topic that is "far away" from the average opinion and $j'$ to a non-controversial topics which is "close" to the average opinion. More concretely, we let $j$ be the topic with $\mathbf{X}_{ij}^{(T)} < \mathbf{X}_{ij}^{(U)}$ that minimizes $|\tau_j - \bar{z}|$; and we let $j'$ be the topic with $\mathbf{X}_{ij'}^{(T)} > \mathbf{X}_{ij'}^{(L)}$ that maximizes $|\tau_{j'} - \bar{z}|$.

*Baseline 2: Strengthening opposing topics* (BL-2). Our second baseline (BL-2) can be viewed as a reverse of the above strategy and is inspired by the experimental outputs that we observed from GDPM: it penalizes non-controversial topics and strengthens topics that are opposing to user $i$'s opinion. More concretely, we compute $\widetilde{z}_{\mathbf{X}^{(T)}}$ and $\bar{z}$ as before. However, then we strengthen the topic $j$ with $\mathbf{X}'_{ij} < \mathbf{X}^{(U)}_{ij}$ that maximizes $-\widetilde{z}_{\mathbf{X}^{(T)}}(i)\tau_j$. For instance, if $\widetilde{z}_{\mathbf{X}^{(T)}}(i) > 0$ then the algorithm will pick the topic $\tau_j < 0$ of largest absolute value; note that since $\widetilde{z}_{\mathbf{X}^{(T)}}$ and $\tau_j$ must have different signs, this corresponds to connecting user $i$ to a topic that opposes its own opinion. Also, we let $j'$ be the topic with $\mathbf{X}'_{ij'} > \mathbf{X}^{(L)}_{ij'}$ and $\widetilde{z}_{\mathbf{X}^{(T)}}(i)\tau_j > 0$ that minimizes $\left|\tau_{j'} - \bar{z}\right|$; this corresponds to our choice of non-controversial topics in BL-1 assuming that $\tau_j$ has the same sign as $\widetilde{z}_{\mathbf{X}^{(T)}}(i)$.

The pseudocode for BL-1 and BL-2 is presented in Algorithm 3 in the appendix.

## 6 EXPERIMENTAL EVALUATION

We evaluate our algorithms on 27 real-world datasets. To conduct realistic experiments, we collect two novel real-world datasets from Twitter, which we denote TwitterSmall ($n = 1\,011$, $m = 1\,960$) and TwitterLarge ($n = 27\,058$, $m = 268\,860$); these two datasets contain ground-truth opinions and we use retweet-information to obtain the aggregate information for the interest- and influencer-matrices $\mathbf{X}$ and $\mathbf{Y}$. We make our novel datasets available in the supplementary material [1] and we will make them public in the non-anonymized version of the paper; we note that TwitterLarge contains more than 27 000 nodes and is thus almost 50 times larger than the previoulsy largest publicly available dataset with ground-truth opinions (which contains less than 550 nodes) [13]. See Appendix B.1 for details on all datasets and on how our new Twitter datasets were collected.

We experimentally compare GDPM against the greedy baselines BL-1 and BL-2. We also compare our gradient-descent algorithm against the black-box solver Convex.jl. In our experiments, given $\mathbf{X}$ and a parameter $\theta \in [0, 1]$, we set $\mathbf{X}^{(U)}_{ij} = \min\{1, \mathbf{X}_{ij} + \theta\}$ and $\mathbf{X}^{(L)}_{ij} = \max\{0, \mathbf{X}_{ij} - \theta\}$, when not mentioned otherwise.

We conduct our experiments on a Linux workstation with a 2.90 GHz Intel Core i7-10700 CPU and 32 GB of RAM. Our code is written in Julia v1.7.2 and available in the supplementary [1].

**Impact of learning rate.** We first study the impact of the learning rate on the convergence of GDPM. Our theoretical analysis suggests using learning rate $L = (8CW/\sqrt{n})\|\mathbf{s}\|_2\|\mathbf{Y}\|_2^2$, which is very large in practice and will result in slow convergence. Thus, we study the convergence of GDPM for different learning rates, in particular, we test $L = 10, 10^2, 10^3, 10^4$. The results for TwitterSmall and TwitterLarge are shown in Figures 2(a) and 2(b), respectively. We observe that even with $L = 10$, GDPM converges to the same objective function value as for much larger values of $L$, and it converges much faster. Therefore, for the rest of our experiments we will use $L = 10$.

**Understanding the behavior of GDPM.** Next, we perform experiments to obtain further insights into which topics are favored by GDPM and which ones are penalized.

To answer this question, we consider the initial interest matrix $\mathbf{X}$ and the matrix $\mathbf{X}^{(T)}$ obtained after GDPM converged. To quantify the behavior of GDPM, we consider the column changes among $\mathbf{X}$ and $\mathbf{X}^{(T)}$. Specifically, for each topic $j$, we measure the change of its weight given by $\delta_j = \sum_i \mathbf{X}^{(T)}_{ij} - \sum_i \mathbf{X}_{ij}$. Note that $\delta_j > 0$ indicates that topic $j$ has more weight in $\mathbf{X}^{(T)}_j$ than in $\mathbf{X}$, i.e., GDPM "favors" it; similarly, $\delta_j < 0$ indicates that topic $j$ has less weight in $\mathbf{X}^{(T)}_j$ than in $\mathbf{X}$, i.e., GDPM "penalizes" it.

In Fig. 1(a) we plot tuples $(\tau_{j,\mathbf{s}}, \delta_j)$ for each topic $j$, where $\delta_j$ is the change in importance for topic $j$, as defined in the previous paragraph, and $\tau_{j,\mathbf{s}} = \sum_{u \in V} \mathbf{Y}_{ju}\mathbf{s}(u)$ is the weighted average of the innate opinions of the influencers for topic $j$. We also color-code the topics based on their content. We observe that GDPM clearly favors topics with large absolute values $|\tau_{j,\mathbf{s}}|$ and it penalizes non-controversial topics with $|\tau_{j,\mathbf{s}}|$ close to 0. We explain this behavior as a consequence of the FJ model opinion dynamics: more controversial topics have a larger impact on the polarization, and to reduce the polarization one has to bring together people from opposing sides.

We note that in all plots, the most favored topics are political. This is surprising, as the algorithm is not aware of the topic labeling. However, we believe this is a consequence of the fact that political topics are among the most controversial (see also below).

In Figures 1(b) and 1(c), we again show the $\delta_j$ values but this time plotted against $\tau_{j,\mathbf{z}_{\mathbf{X}}}$ using the original expressed opinions $\mathbf{z}_{\mathbf{X}}$ (before optimization) and $\tau_{j,\mathbf{z}_{\mathbf{X}^{(T)}}}$ using the final expressed opinions $\mathbf{z}_{\mathbf{X}^{(T)}}$ (after optimization). Qualitatively, we observe the same behavior as before, so that more controversial topics are favored and non-controversial topics are penalized. Observe that now the $x$-axes have smaller scales, since the expressed opinions are contractions of the innate opinions. Here, it is important to observe that before the optimization (Fig. 1(b)) the average topic opinions were in $[-0.066, 0.118]$ and after the optimization (Fig. 1(c)) they are in $[-0.028, 0.076]$. Thus, the algorithm clearly brought all topics closer together.

Next, we study the behavior of GDPM when we do not allow to make any changes on the accounts' interests in political topics, i.e., we set $\mathbf{X}^{(U)}_{ij} = \mathbf{X}_{ij}$ and $\mathbf{X}^{(L)}_{ij} = \mathbf{X}_{ij}$ for all political topics $j$ and all accounts $i$. In Fig. 1(d)–(f) we show the same plots as in Fig. 1(a)–(c), when weight changes for political topics are not allowed. We obtain the same qualitative outcome as before: controversial topics are favored and non-controversial topics are penalized. We use these qualitative insights to develop the second baseline algorithm BL-2.

As expected, when weight changes for political topics are not allowed, we obtain a restricted version of the problem which limits the disagreement-polarization reduction. For reference, in the setting of Fig. 1(a)–(c), when weight changes for all topics are allowed, the disagreement-polarization index is reduced to 93.44% of its original value. In contrast, in the setting of Fig. 1(d)–(f), with no changes on political topics, the disagreement-polarization index is reduced to only 97.69% of its original value.

We stress that the above fine-grained analysis of which topics are penalized and favored in the FJ model has only become possible due to the introduction of our model from Section 4.

**Comparison with black-box convex solver.** Since Problem 3 is convex, we compare our gradient-descent based algorithm GDPM

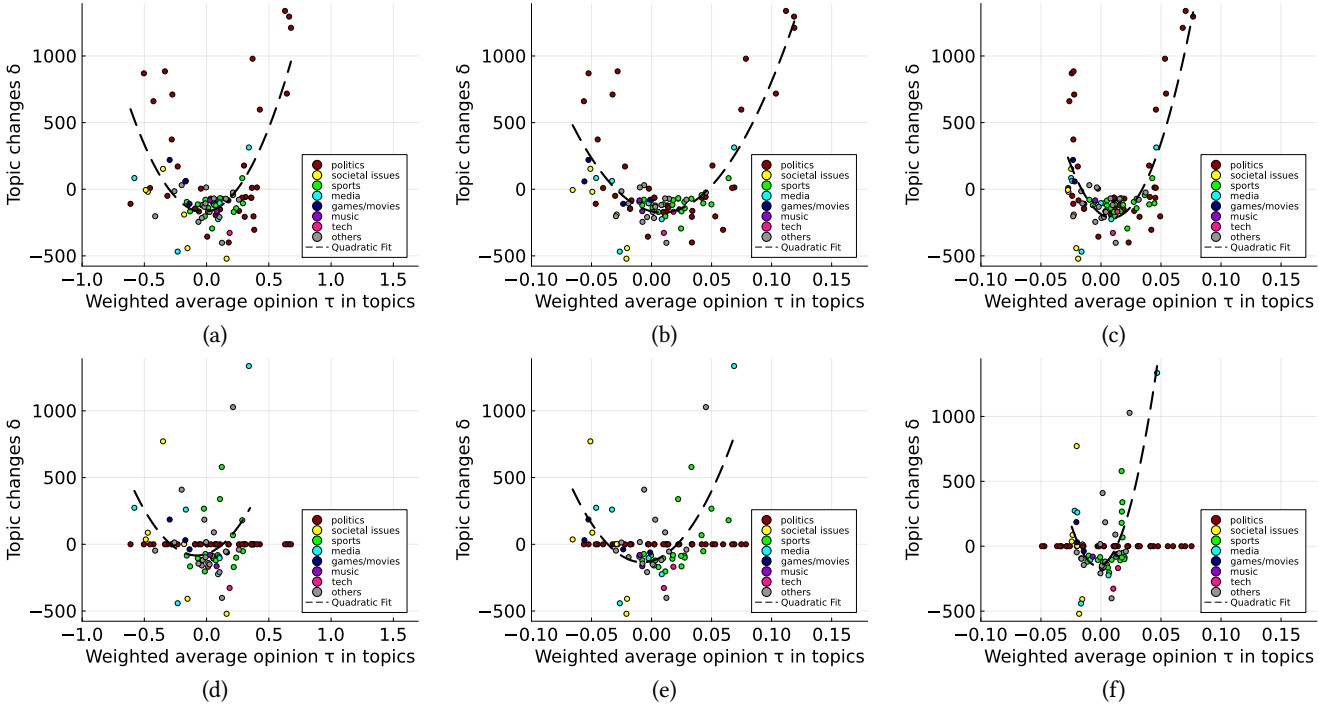

Figure 1: **Behavior of GDPM on TwitterLarge** ($\theta = 0.1$, $C = 0.1$, $L = 10$). **We report the change of topic importance ($y$-axis) and the weighted average of the opinions of influential users for each topic ($x$-axis): (a) weighted innate opinions $\tau_{j,s}$; (b) weighted expressed opinions before optimization $\tau_j$; (c) weighted expressed opinions after optimization $\tau_j$. (d)—(f) repeat the same plots when the algorithm must not change interest in political topics. For reference, the results are fitted with a quadratic function.**

Table 2: **Comparison of reduction ratio $\frac{f(\mathbf{X}^{(T)})}{f(\mathbf{X})}$(%) of GDPM, BL-1, and BL-2 on real-world graphs. In the experiments we set $\theta = 0.1$ and $C = 0.1$. We used $L = 10$ and $T = 100$ for GDPM and we set $T = 10$ for the greedy baselines.**

| Graph | GDPM | BL-1 | BL-2 | Graph | GDPM | BL-1 | BL-2 |
|---|---|---|---|---|---|---|---|
| Erdos992 | **94.34** | 100 | 94.97 | Themarker | **85.62** | 100 | 89.30 |
| Advogato | **91.36** | 100 | 92.59 | Slashdot | **92.23** | 100 | 93.43 |
| PagesGovernment | **87.82** | 100 | 88.62 | BlogCatalog | **85.59** | 100 | 89.77 |
| WikiElec | **87.30** | 100 | 89.72 | WikiTalk | **92.82** | 100 | 93.47 |
| HepPh | **86.13** | 100 | 88.69 | Gowalla | **91.79** | 100 | 92.63 |
| Anybeat | **92.17** | 100 | 93.21 | Academia | **92.04** | 100 | 93.37 |
| PagesCompany | **92.41** | 100 | 93.28 | GooglePlus | **86.43** | 100 | 88.28 |
| AstroPh | **88.20** | 100 | 89.74 | Citeseer | **90.53** | 100 | 91.48 |
| CondMat | **91.75** | 100 | 94.36 | MathSciNet | **93.55** | 100 | 93.93 |
| Gplus | **93.91** | 100 | 94.48 | TwitterFollows | **94.22** | 100 | 95.59 |
| Brightkite | **93.02** | 100 | 93.99 | YoutubeSnap | **93.58** | 100 | 94.76 |

against Convex.jl. Convex.jl is a popular black-box convex optimization tool written in Julia. Our experiments show that GDPM is orders of magnitude more efficient than Convex.jl. In particular, even though for this experiment we used 102 GB of RAM, running Convex.jl on graphs with more than 500 nodes exceeds the memory constraint. In contrast, GDPM scales up to graphs with millions of nodes and edges (see Tables 3 and 2). For the details of these experiments, see Section B.2. Here, one of the bottlenecks for Convex.jl is that it cannot access our efficient opinion estimation routine from

Proposition 3. In Table 3 we show the routine from the proposition is indeed orders of magnitude more efficient than estimating the opinions using naïve matrix inversion.

**Comparison with greedy baselines and varying parameters.** Next, we compare GDPM against the baselines BL-1 and BL-2, and vary the parameters $\theta$ and $C$. Note that since GDPM is guaranteed to converge to an optimal solution, we expect it to outperform both baselines.

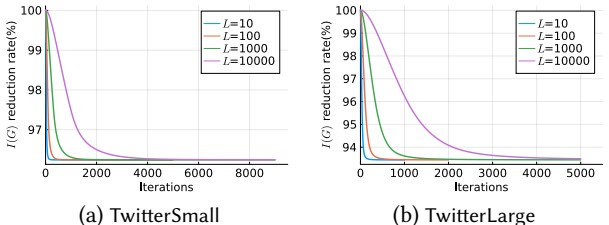

(a) TwitterSmall

(b) TwitterLarge

**Figure 2: Convergence of GDPM for different learning rates on two Twitter datasets ($\theta = 0.1$, $C = 0.1$). The $y$-axis shows the reduction ratio $f(X_{ALG})/f(X)$.**

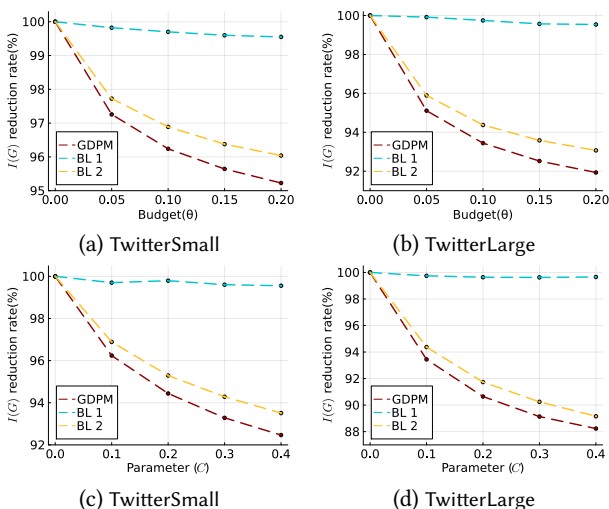

(a) TwitterSmall

(b) TwitterLarge

(c) TwitterSmall

(d) TwitterLarge

**Figure 3: Reduction of the disagreement–polarization index on two datasets for all of our algorithms ($L = 10$). The $y$-axis shows the reduction ratio $f(X_{ALG})/f(X)$. In (a)-(b) we set $C = 0.1$ and vary $\theta \in \{0.05, 0.1, 0.15, 0.2\}$. In (c)-(d) we set $\theta = 0.1$ and vary $C \in \{0.1, 0.2, 0.3, 0.4\}$.**

We report the results of all algorithms with varying $\theta \in \{0.05, 0.1, 0.15, 0.2\}$ in Figures 3(a)–(b) for TwitterSmall and TwitterLarge. As expected, GDPM obtains the largest reduction of the objective. Furthermore, BL-2 outperforms BL-1 by a large margin. This is not surprising, since we design BL-2 based on the insights that we get from analyzing the behavior of GDPM; the observed behavior thus suggests that our intuition about GDPM is correct. In addition, we observe that the reduction in disagreement and polarization increases with $\theta$. This behavior is also well-aligned with our expectation, as larger values of $\theta$ enlarge the feasible space and allow for more flexibility in recommending interesting topics to all Twitter accounts.

In addition, in Figures 3(c)–(d) we report the results of all algorithms with varying $C \in \{0.1, 0.2, 0.3, 0.4\}$ for TwitterSmall and TwitterLarge. The behavior of all algorithms remains consistent: GDPM achieves the largest reduction, while BL-2 outperforms BL-1. As expected, the reduction in disagreement and polarization increases with $C$, since larger values of $C$ allow for more impact of the timeline algorithm.

Finally, we note that GDPM achieves a larger reduction on TwitterLarge than on TwitterSmall throughout all experiments. This is perhaps a bit surprising since on both datasets we increase the total edge weight by a $C$-fraction. However, the average node degree of TwitterLarge is larger than for TwitterSmall. Furthermore, the user–topic matrix $X$ and influenc–topic matrix $Y$ have different structure for TwitterSmall and TwitterLarge, which results in the low-rank adjacency matrix $A_X$ containing 25% and 33% of non-zero entries, respectively. We believe that both of these characteristics of the datasets lead to higher connectivity in TwitterLarge, which results in better averaging of the opinions and thus ultimately in less polarization and disagreement.

**Performance of the optimization algorithms.** We report the optimization results of GDPM, BL-1, and BL-2 in Table 2. We used different real-world graphs with synthetically generated polarized opinions and synethically generated matrices $X$ and $Y$. We run the greedy baselines 10 iterations due to the high computation cost and choose the best $X^{(T)}$ as output in our experiments.

We observe that GDPM outperforms the two baselines on all graphs. This is the expected behavior, since GDPM guarantees decreasing the objective function constantly and converges to the optimal solution; the two baselines have no such property. Across all datasets, GDPM decreases the polarization and disagreement by at least 5.6% and by up to 14.4%. Furthermore, BL-2 outperforms BL-1 by a large margin and its results are often not much worse than those of GDPM. Interestingly, BL-1 cannot reduce the polarization and disagreement on all graphs.

We include additional experiments, including a running-time analysis, in the appendix.

# 7 CONCLUSION

We showed how to augment the popular FJ model to take into account aggregate information of timeline algorithms. This allows us to bridge between network-level opinion dynamics and user-level recommendations. For our model, we presented an algorithm that provably approximates the measures of polarization and disagreement in near-linear time. We also considered the problem of optimizing the timeline algorithm, so as to minimize polarization and disagreement in the network, and developed an efficient gradient-descent algorithm, GDPM, which computes an $(1 + \epsilon)$-approximate solution in time $\widetilde{O}(m\sqrt{n}\log(1/\epsilon))$ under realistic parameter settings. Our experiments confirm the efficiency and effectiveness of the proposed methods and showed that our gradient-descent algorithm is orders of magnitude faster than a black-box solver. We also released the largest graph datasets with ground-truth opinions.

We believe that our work provides several directions for future research. First, extensions to the non-symmetric setting are highly interesting. Second, it will be valuable to consider other opinion-formation models, beyond the FJ model, and compare the results. Third, it will be intriguing to design more complex models, capturing real-world nuances, that allow us to bridge between opinion dynamics and properties of present-day timeline algorithms.

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

## A OMITTED PSEUDOCODE

In this section, we present the pseudocode of Algorithms 1, 2 and 3.

---

**Algorithm 1:** Compute an approximation $\widetilde{z}_X$ of $z_X$

**Input:** Innate opinion $s$, user–topic matrix $X$, influence–topic matrix $Y$, fraction of *weight* $C$, error parameter $\epsilon$

**Output:** Approximated expressed opinion $\widetilde{z}_X$

1 $\epsilon_{z_1} = \frac{\epsilon}{4} \cdot \min \left\{ 1, \frac{2n}{200 \cdot CW \cdot \|U\|_2 \cdot \|V\|_2} \right\}$

2 $\epsilon_R = \frac{1}{2k} \min \left\{ 0.009 \frac{2n}{CW \cdot \|V\|_F}, \frac{2n}{10^5 \cdot CW \cdot \|V\|_2} \cdot \min \left\{ 100, \frac{2n}{CW} \cdot \frac{\epsilon/4}{\|U\|_2 \cdot \|V\|_2 \cdot \|s\|_2} \right\} \right\}$

3 $\epsilon_{z_2} = \frac{2n}{CW} \cdot \frac{\epsilon}{4}$

4 $M \leftarrow I + L + \text{diag}(A_X 1)$

5 $U \leftarrow \begin{pmatrix} X & Y^\top \end{pmatrix}, V \leftarrow \begin{pmatrix} Y \\ X^\top \end{pmatrix}$

6 $z_1 \leftarrow \text{Solve}(M, s, \epsilon_{z_1})$

7 $y_1 \leftarrow V z_1$

8 $R \leftarrow$ the $n \times (2k)$ matrix, where the $j$-th column is given by $\text{Solve}(M, w_j, \epsilon_R)$ with $w_j$ denoting the $j$-th column of $U$ for all $j$

9 $S \leftarrow I - \frac{CW}{2n} VR$

10 $T \leftarrow S^{-1}$

11 $y_2 \leftarrow T y_1$

12 $y_3 \leftarrow U y_2$

13 $z_2 \leftarrow \text{Solve}(M, y_3, \epsilon_{z_2})$

14 **return** $\widetilde{z}_X \leftarrow z_1 + \frac{CW}{2n} z_2$

---

**Algorithm 2:** GDPM

**Input:** Innate opinion $s$, user–topic matrix $X$, influence–topic matrix $Y$, budget $\theta$, fraction of *weight* $C$

**Output:** User–topic matrix $X^{(T)}$ after optimization

1 $L \leftarrow \frac{8CW}{\sqrt{n}} \cdot \|s\|_2 \cdot \|Y\|_2^2$

2 $X^{(0)} \leftarrow X$

3 **for** $T = 1, \ldots, O\left( \sqrt{\frac{CWkn}{\epsilon}} \right)$ **do**

4      Compute $\widetilde{z}_{X^{(T)}}$ using Algorithm 1

5      $\widetilde{\nabla}_X f(X^{(T)}) \leftarrow \frac{CW}{2n}(2 \cdot \widetilde{z}_{X^{(T)}} \cdot \widetilde{z}_{X^{(T)}}^\top \cdot Y^\top - \widetilde{z}_{X^{(T)}} \odot \widetilde{z}_{X^{(T)}} \cdot 1_k^\top - 1_n \cdot (\widetilde{z}_{X^{(T)}}^\top \odot \widetilde{z}_{X^{(T)}}^\top) \cdot Y^\top)$

6      $V^{(T)} \leftarrow$ the matrix where the $i$-th row is given by $\text{Proj}_Q\left( X_i^{(T)} - \frac{1}{L}(\widetilde{\nabla}_X f(X^{(T)}))_i \right)$

7      $\alpha_T \leftarrow \frac{T+1}{2}$

8      $W^{(T)} \leftarrow$ the matrix where the $i$-th row is given by $\text{Proj}_Q\left( (X^{(0)})_i - \frac{1}{2L} \sum_{t=1}^{T} \alpha_t (\widetilde{\nabla}_X f(X^{(t)}))_i \right)$

9      $A_T \leftarrow \sum_{i=0}^{T} \alpha_i$

10      $\tau_T \leftarrow \frac{\alpha_T}{A_T}$

11      $X^{(T+1)} \leftarrow \tau_T V^{(T)} + (1 - \tau_T) W^{(T)}$

12 **return** $X^{(T)}$

---

**Algorithm 3:** Baselines BL-1 and BL-2

---

**Input:** Innate opinion $\mathbf{s}$, user–topic matrix $\mathbf{X}$, influence–topic matrix $\mathbf{Y}$, lower-bound matrix $\mathbf{X}^{(L)}$, upper-bound matrix $\mathbf{X}^{(U)}$, maximum iterations $T_{max}$

**Output:** User–topic matrix $\mathbf{X}^{(T)}$ after optimization

1   $\mathbf{X}^{(0)} \leftarrow \mathbf{X}$

2   **for** $T = 1, \ldots, T_{\max}$ **do**

3     $\mathbf{X}^{(T)} \leftarrow \mathbf{X}^{(T-1)}$

4     Compute $\widetilde{\mathbf{z}}_{\mathbf{X}^{(T)}}$ using Algorithm 1

5     $\bar{z} = \frac{1}{n} \sum_{u \in V} \widetilde{\mathbf{z}}_{\mathbf{X}^{(T)}}(u)$

6     $\tau_j = \sum_{u \in V} \mathbf{Y}_{ju} \widetilde{\mathbf{z}}_{\mathbf{X}^{(T)}}(u)$

7     **for** *each row $i$* **do**

8       **if** *(BL-1)* **then**

9         $j \leftarrow$ topic with $\mathbf{X}_{ij}^{(T)} < \mathbf{X}_{ij}^{(U)}$ minimizing $\left|\tau_j - \bar{z}\right|$

10         $j' \leftarrow$ topic with $\mathbf{X}_{ij'}^{(T)} > \mathbf{X}_{ij'}^{(L)}$ minimizing $\left|\tau_{j'} - \bar{z}\right|$

11       **if** *(BL-2)* **then**

12         $j \leftarrow$ topic with $\mathbf{X}'_{ij} < \mathbf{X}_{ij}^{(U)}$ minimizing $-\widetilde{\mathbf{z}}_{\mathbf{X}^{(T)}}(i)\tau_j$

13         $j' \leftarrow$ topic with $\mathbf{X}'_{ij'} > \mathbf{X}_{ij'}^{(L)}$ and $\widetilde{\mathbf{z}}_{\mathbf{X}^{(T)}}(i)\tau_j > 0$ minimizing $\left|\tau_{j'} - \bar{z}\right|$

14       $\delta \leftarrow \min\{\mathbf{X}_{ij}^{(U)} - \mathbf{X}_{ij}^{(T)}, \mathbf{X}_{ij'}^{(T)} - \mathbf{X}_{ij'}^{(L)}\}$

15       $\mathbf{X}_{ij}^{(T)} \leftarrow \mathbf{X}_{ij}^{(T)} + \delta$

16       $\mathbf{X}_{ij'}^{(T)} \leftarrow \mathbf{X}_{ij'}^{(T)} - \delta$

17 **return** $\mathbf{X}^{(T)}$

---

# B   OMITTED EXPERIMENTS

## B.1   Data Collection and Parameter Settings

**Datasets.** We begin by describing our data-collection process. Starting from a list of Twitter accounts who actively engage in political discussions in the US, which was compiled by Garimella and Weber [18], we randomly sample two smaller subsets of 5 000 and 50 000 accounts, respectively. Since the dataset was more than 6 years old, only approximately 30-50% of the accounts are still active or publicly accessible. For these accounts, we obtained the entire list of followers, except for users with more than 100 000 followers for whom we got only the 100 000 most recent followers (users with more than 100 000 followers account for less than 2% of our dataset). We also obtained the last 3 200 tweets they posted on their own timeline. We use multiple Twitter-API keys and parallelize the data collection. The data collection was started in March 2022 and took over a week to finish.

Based on this obtained information, we construct two graphs in which the nodes correspond to Twitter accounts and the edges correspond to the accounts' following relationships. Then we consider only the largest connected component in each network and denote the resulting datasets TwitterSmall and TwitterLarge respectively. In the end, TwitterSmall contains 1 011 nodes and 1 960 edges. TwitterLarge contains 27 058 nodes and 268 860 edges.

To obtain the innate opinions of the nodes in the graphs, we proceed as follows. First, we compute the political polarity score for each account using the method proposed by Barberá [4], which has been used widely in the literature [7, 8]. The polarity scores range from -2 to 2 and are computed based on following known political accounts. To obtain the innate opinions $\mathbf{s}$ of the retrieved accounts, we center the political scores to 0 and rescale them into the interval $[-1, 1]$. We visualize the innate opinions of the accounts of TwitterSmall in Fig. 4(a) and of TwitterLarge in Fig. 4(b). We observe that the distribution of the opinions is relatively similar in both datasets, and that the opinion scores are significantly polarized. A plausible explanation of this phenomenon is that our seed set consists of politically active accounts in the US, which are more likely to support one of the two extremes of the political spectrum than having moderate opinions.

**User–topic and influence–topic matrices.** Next, we explain how we obtained the user–topic matrix $\mathbf{X}$ and the influence–topic matrix $\mathbf{Y}$. We note that in an academic environment, it is impossible to obtain these matrices exactly, since we cannot obtain data on how the timelines of different users are composed and how the posts for each topic are picked by the timeline algorithms that are deployed by online social networks. Therefore, we obtain $\mathbf{X}$ and $\mathbf{Y}$ by using retweet-data as a surrogate, which indicates the users' interest and impact on different topics. We now describe this process in detail.

We use textual information and hashtags in the tweets dataset to estimate the interest of accounts and influential accounts in different topics. More concretely, we start by finding all hashtags that are used in the historical tweets, and collect the hashtags used by each account. We then apply tf-idf on this data, where the documents correspond to accounts and the terms correspond to hashtags. The result gives a matrix $\mathbf{B}$, in which each entry $\mathbf{B}_{uv}$ corresponds to the tf-idf score of account $u$ for hashtag $v$. Next, we apply non-negative matrix

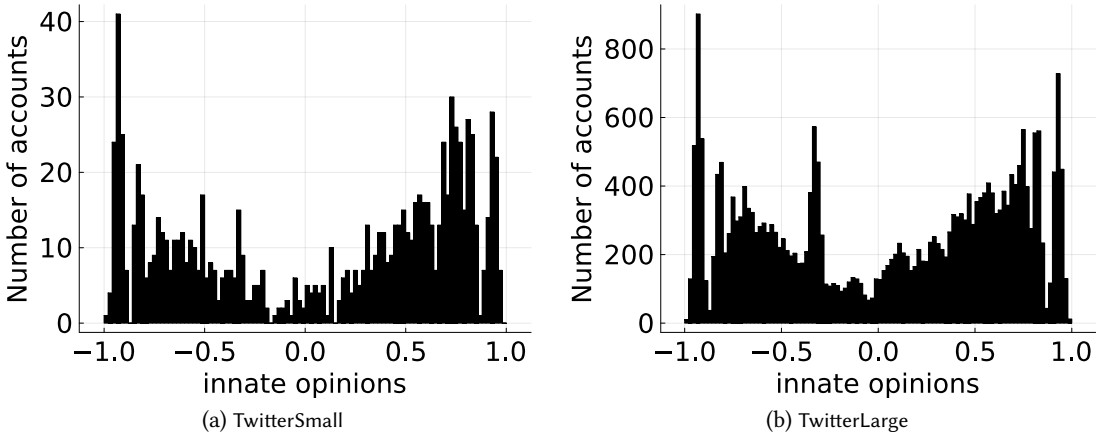

(a) TwitterSmall

(b) TwitterLarge

**Figure 4: Innate opinion distributions on our two real-world Twitter datasets.**

factorization (NMF) on **B** to obtain topics from this matrix. NMF on a `tf-idf` matrix has been shown to produce coherent topics in the past [23]. The NMF procedure produces two matrices: $\mathbf{W} \in \mathbb{R}^{n \times k}$ and $\mathbf{H} \in \mathbb{R}^{k \times h}$, such that $\mathbf{B} \approx \mathbf{WH}$. Here, $n$ is the number of accounts in the dataset, $h$ is the number of distinct hashtags, and the latent dimension $k$ is the number of topics that we wish to find. We systematically test different values of $k$ from 50–100 and find that for our data, $k = 100$ produces the most reasonable topics. Therefore, in our experiments we use $k = 100$.

By the semantics of matrix factors in NMF, we interpret $\mathbf{W}_{ij}$ as an indicator of the interest of account $i$ in topic $j$. Therefore, we set the $i$-th row of the interest matrix **X** to $\mathbf{X}_i = \mathbf{W}_i / \sum_{j=1}^{k} \mathbf{W}_{ij}$, so as to satisfy the row-stochastic constraint, i.e., $\sum_j \mathbf{X}_{ij} = 1$, for all $i$.

Similarly, we interpret $\mathbf{H}_{jh}$ as the importance of hashtag $h$ in topic $j$. To avoid using hashtags that are too noisy, we consider only the most frequent hashtags that make up for the 60% of the volume of all hashtags. We then set the influence–topic matrix **Y** to the percentage of retweets that an account receives for each topic. More concretely, we let $r_{ih}$ denote the number of retweets for tweets posted by account $i$ that contain hashtag $h$. We set $\mathbf{Y}'$ to the matrix with $\mathbf{Y}'_{ji} = \sum_{h \in S_j} r_{ih}$, i.e., $\mathbf{Y}'_{ji}$ is the number of retweets for tweets posted by account $i$ that contain hashtags assigned to topic $j$. We then compute **Y** by normalizing the rows of $\mathbf{Y}'$, i.e., we set $\mathbf{Y}_{ji} = \mathbf{Y}'_{ji} / \sum_{i=1}^{n} \mathbf{Y}'_{ji}$ to ensure that **Y** is row stochastic.

**Upper and lower bounds $\mathbf{X}^{(U)}$ and $\mathbf{X}^{(L)}$.** Given a matrix **X** and a parameter $\theta \in [0, 1]$, in our experiments (unless mentioned otherwise) we construct the element-wise upper-bound matrix $\mathbf{X}^{(U)}$ and the lower-bound matrix $\mathbf{X}^{(L)}$ by setting $\mathbf{X}^{(U)}_{ij} = \min\{1, \mathbf{X}_{ij} + \theta\}$ and $\mathbf{X}^{(L)}_{ij} = \max\{0, \mathbf{X}_{ij} - \theta\}$. Intuitively, we can consider $\theta$ as a budget that the algorithm has to redistribute for each entry of **X**. Note, however, that in the presence of topic label information, we can set topic-specific bounds. For example, if we set $\mathbf{X}^{(U)}_{ij} = \mathbf{X}_{ij}$, we then forbid the algorithm to increase account $i$'s interest in topic $j$. We apply this idea in some of our experiments, by setting different bounds for political topics. See Figures 1 (d)–(f) for details.

**Additional datasets.** To compare our algorithms across more datasets, we also consider several real-world graphs for which we synthetically generate the innate opinions, the user–topic matrix **X**, and the influenc–topic matrix **Y**.

The real-world graphs that we consider are publicly available from the Network Repository [35]. Our experiments were conducted on the largest connected component of each dataset. Table 3 lists the networks that we consider in increasing order of the number of nodes. The largest network has more than two million nodes, while the smallest one has 4 991 nodes.

Next, we consider four distributions to generate the innate opinions: uniform, power-law, exponential, and a custom "polarized" distribution. For the first three distributions, we use the same parameter setting as Xu et al. [38]. Note that they compute innate opinion $s \in [0, 1]$ and here we rescale the innate opinions to $[-1, 1]$. In the "polarized" distribution, we mimic the opinion distribution from TwitterSmall and TwitterLarge in Figure 4, where the innate opinions tend to be concentrated at the two opposite extremes, while sparsely distributed around the middle. Thus, here we generate "polarized" opinions as follows. For each node $i$, we generate a value $x_i$ based on the exponential opinion distribution from above. Now for the first $n/2$ nodes we set their innate opinion to $\mathbf{s}_i = x_i$ and for the remaining $n/2$ nodes we set their opinion to $\mathbf{s}_i = 1 - x_i$. Then we rescale **s** such that all opinion are in $[-1, 1]$.

We also compute synthetic user–topic matrices **X** and influence–topic matrices **Y** by simulating properties of TwitterSmall and TwitterLarge. More concretely, for TwitterLarge we visualize the distribution of elements in **X** and **Y** in Fig. 5. It shows that the entries in **X** and **Y** follow a power-law distribution.

To generate **X** we proceed as follows. For each row $\mathbf{X}_i$ that we generate synthetically, we sample the entries $\mathbf{X}_{ij}$ from a power-law distribution with $\alpha = 2.5$ (this value of $\alpha$ was also used in [38]). We control the sparsity of the matrix by removing elements with a value

Table 3: Results of Algorithm 1 for computing $\widetilde{z}_X$ on real-world graphs. We report graph statistics, comparison of running times (in seconds) and approximation errors for computing $\widetilde{z}_X$ *exactly* and computing $\widetilde{z}_X$ *approximately* using Algorithm 1. We use four innate opinion distributions (uniform, power-law, exponential, and a custom "polarized" distribution). The synthetic user–topic and influence–topic matrices X and Y were drawn from the distributions described in the text. We set $C = 0.1$. This experiment was conducted in a Linux server with E5-2630 V4 processor (2.2 GHz) and 128 GB memory.

| Graph | $n$ | $m$ | Uniform | | | Power-law | | | Exponential | | | Polarized | | |
|---|---|---|---|---|---|---|---|---|---|---|---|---|---|---|
| | | | Exact | Approx | Error | Exact | Approx | Error | Exact | Approx | Error | Exact | Approx | Error |
| Erdos992 | 4,991 | 7,428 | 10.08 | 0.58 | 0.0108 | 9.57 | 0.49 | 0.0794 | 9.71 | 0.51 | 0.0794 | 9.71 | 0.51 | 0.0876 |
| Advogato | 5,054 | 39,374 | 10.26 | 1.02 | 0.0180 | 10.26 | 1.12 | 0.1134 | 10.16 | 1.15 | 0.1134 | 10.16 | 1.15 | 0.2462 |
| PagesGovernment | 7,057 | 89,429 | 26.17 | 1.76 | 0.0040 | 25.72 | 1.77 | 0.1553 | 25.94 | 1.71 | 0.1553 | 25.94 | 1.71 | 0.0597 |
| WikiElec | 7,066 | 100,727 | 26.41 | 1.78 | 0.0057 | 26.12 | 1.53 | 0.0064 | 26.12 | 1.62 | 0.0064 | 26.12 | 1.62 | 1.0133 |
| HepPh | 11,204 | 117,619 | 98.00 | 2.56 | 0.0095 | 97.89 | 2.73 | 0.1245 | 98.04 | 2.61 | 0.1245 | 98.04 | 2.61 | 0.0138 |
| Anybeat | 12,645 | 49,132 | 140.91 | 1.82 | 0.0010 | 140.96 | 1.87 | 0.0132 | 141.75 | 2.07 | 0.0132 | 141.75 | 2.07 | 0.0073 |
| PagesCompany | 14,113 | 52,126 | 195.51 | 2.45 | 0.0099 | 195.93 | 2.52 | 0.0154 | 194.27 | 2.46 | 0.0154 | 194.27 | 2.46 | 0.0473 |
| AstroPh | 17,903 | 196,972 | 400.09 | 4.53 | 0.0282 | 398.54 | 4.86 | 0.0017 | 402.21 | 4.78 | 0.0017 | 402.21 | 4.78 | 0.0075 |
| CondMat | 21,363 | 91,286 | 674.03 | 4.04 | 0.0011 | 669.62 | 4.05 | 0.1119 | 672.78 | 4.28 | 0.1119 | 672.78 | 4.28 | 0.0189 |
| Gplus | 23,613 | 39,182 | 902.86 | 3.06 | 0.0002 | 919.68 | 2.62 | 0.0047 | 905.09 | 2.59 | 0.0047 | 905.09 | 2.59 | 0.0102 |
| Brightkite | 56,739 | 212,945 | 13864.33 | 14.71 | 0.0007 | 13366.60 | 14.04 | 0.0119 | 14012.49 | 16.03 | 0.0119 | 14012.49 | 16.03 | 0.0720 |
| Themarker | 69,317 | 1,644,794 | — | 37.06 | — | — | 36.34 | — | — | 37.81 | — | — | 36.84 | — |
| Slashdot | 70,068 | 358,647 | — | 16.01 | — | — | 14.76 | — | — | 14.45 | — | — | 14.47 | — |
| BlogCatalog | 88,784 | 2,093,195 | — | 43.20 | — | — | 41.67 | — | — | 43.90 | — | — | 43.39 | — |
| WikiTalk | 92,117 | 360,767 | — | 16.53 | — | — | 16.23 | — | — | 15.89 | — | — | 16.78 | — |
| Gowalla | 196,591 | 950,327 | — | 56.87 | — | — | 51.58 | — | — | 53.04 | — | — | 53.13 | — |
| Academia | 200,167 | 1,022,440 | — | 63.00 | — | — | 63.95 | — | — | 60.47 | — | — | 61.90 | — |
| GooglePlus | 201,949 | 1,133,956 | — | 47.89 | — | — | 48.57 | — | — | 47.38 | — | — | 48.10 | — |
| Citeseer | 227,320 | 814,134 | — | 46.57 | — | — | 47.23 | — | — | 46.45 | — | — | 46.76 | — |
| MathSciNet | 332,689 | 820,644 | — | 68.91 | — | — | 62.26 | — | — | 67.23 | — | — | 61.37 | — |
| TwitterFollows | 404,719 | 713,319 | — | 44.10 | — | — | 41.91 | — | — | 42.19 | — | — | 43.16 | — |
| Delicious | 536,108 | 1,365,961 | — | 108.95 | — | — | 112.19 | — | — | 115.72 | — | — | 126.50 | — |
| YoutubeSnap | 1,134,890 | 2,987,624 | — | 273.09 | — | — | 271.67 | — | — | 262.28 | — | — | 262.05 | — |
| Flickr-und | 1,624,992 | 15,476,835 | — | 858.09 | — | — | 857.98 | — | — | 858.73 | — | — | 904.29 | — |
| Flixster | 2,523,386 | 7,918,801 | — | 663.40 | — | — | 674.12 | — | — | 644.32 | — | — | 653.05 | — |

The running-time header reads: Running time (s) of evaluating $z_X$ (Exact) and $\overline{z}_X$ (Approx) with Algorithm 1, and approximation error (Error, $\times 10^{-8}$)

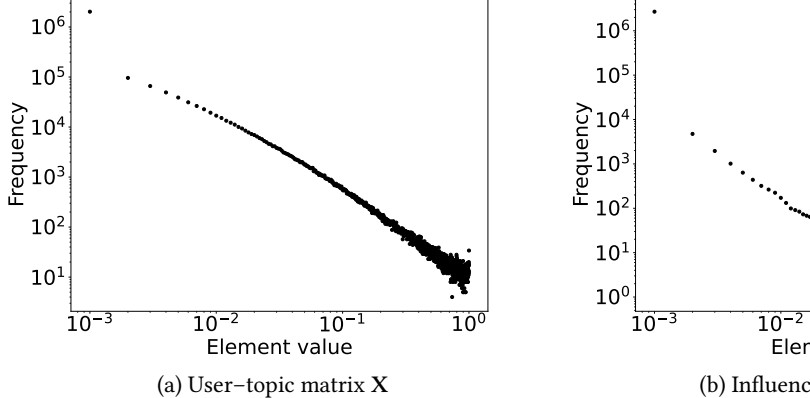

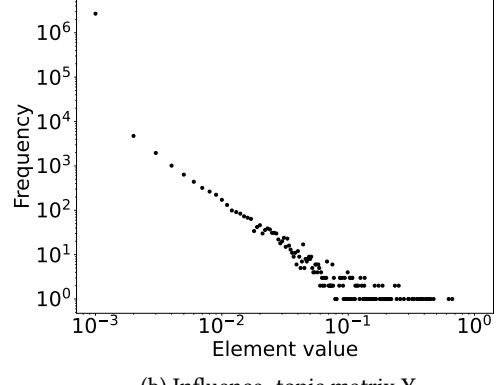

(a) User–topic matrix X      (b) Influence–topic matrix Y

**Figure 5: Distribution of elements in X and Y in TwitterLarge.**

smaller than 0.25 and rescaling the result such that $X_i$ is row-stochastic; we chose the value 0.25 to match the sparsity of of our real-world matrices from TwitterLarge.

To generate Y, we first observe from Fig. 1 that $\tau_{j,\mathbf{s}}$, the weighted average of the innate opinions for each topic $j$, ranges from -0.65 to 0.65 and the majority of topics are located around 0. Inspired by this fact, we construct the topic–influence matrix Y such that the value $\tau_{j,\mathbf{s}}$ are spread across the opinion spectrum (similar to the real-world behavior). More concretely, we equally divide the opinion spectrum $[-1, 1]$ into $d$ chunks and we assign a weight $w_i$ to each chunk $i$. Now, for each topic $j$ we first sample its *bias*. That is, we sample a chunk $i$ with probability proportional to the weight $w_i$ and then all users of topic $j$ have their innate opinion from chunk $i$. If $V_i$ denotes the set of users with innate opinion in chunk $i$ and $n$ is the number of all users, then we pick $0.02n$ users from $V_i$ uniformly at random and for each $u \in V_i$, we set $Y_{ju}$ using a power-law distribution with $\alpha = 2.5$. Finally, we rescale the result such that $Y_i$ is row-stochastic. In our synthetic experiments we used $d = 3$ and $w = [0.3, 0.4, 0.3]$.

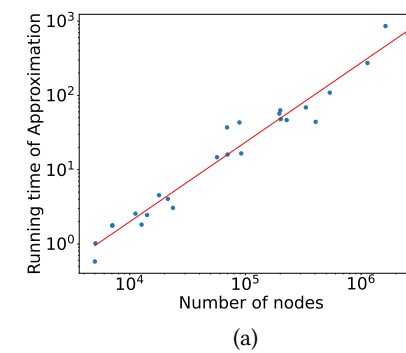
(a)

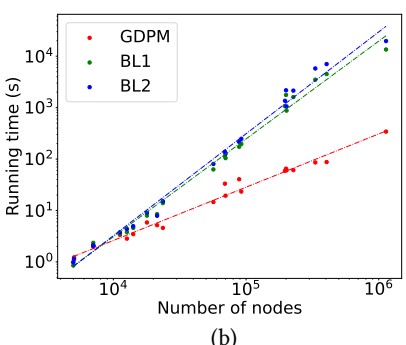
(b)

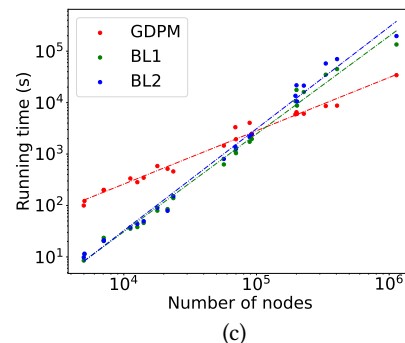
(c)

**Figure 6: Running time of algorithms in seconds. Each marker corresponds to one of the datasets from Table 3. We fit a linear regression to indicate trends. Plot (a) shows the running time of Algorithm 1 for computing $\widetilde{z}_X$. Plot (b) shows the running time for a single iteration of GDPM, BL-1 and BL-2. Plot (c) shows the total running times for 100 iterations of GDPM and 10 iterations of BL-1 and BL-2.**

**Table 4: Running times of the algorithms. We report the time (in seconds) required for a single iteration with _approximate_ expressed opinions $\widetilde{z}_X$ and with _exact_ expressed opinions $z_X$. We also report the _total_ running time over 150 iterations when using the approximate solver. Here, we set $L = 10$, $C = 0.1$, $\theta = 0.1$ and $T = 150$.**

| Algorithm | TwitterSmall | | | TwitterLarge | | |
|---|---|---|---|---|---|---|
| | **Approx** (1 iter.) | **Exact** (1 iter.) | **Total** (150 iter.) | **Approx** (1 iter.) | **Exact** (1 iter.) | **Total** (150 iter.) |
| **GDPM** | 0.21 | 0.15 | 31.95 | 7.67 | 1530.02 | 1150.06 |
| **BL 1** | 0.09 | 0.12 | 13.51 | **4.76** | 1501.21 | **713.97** |
| **BL 2** | **0.08** | 0.13 | **12.67** | 4.98 | 1485.77 | 747.07 |

We note that this way of $Y$ was crucial to obtain our experimental results on synthetic data. Initially, we simply picked $0.02n$ users for each topic uniformly at random. However, this resulted in all $\tau_{j,s}$ being very close to 0, which is not the behavior that we saw in our real-world datasets. This also had the side-effect hat our optimization algorithms could not reduce the polarization and disagreement significantly.

## B.2 Additional experiment results

Now we report additional experimental results, including a running time analysis.

**Approximating Expressed Opinions $\widetilde{z}_X$.** Table 3 reports running time and approximation error of Algorithm 1 for computing $\widetilde{z}_X$ on different real-world graphs. We compare against the exact solution $z_X$ and note that we cannot compute $z_X$ for the 14 largest graphs due to the high running time of computing the exact solution. We observe that Algorithm 1 is orders of magnitude faster than the naïve computation of $z_X$ and its error is negliglible in practice (note that errors are typically less than $10^{-8}$).

We also visualize the running times from Table 3 for uniformly distributed innate opinions in Fig. 6(a). We observe that the running time grows linearly with the number of nodes.

We also note that in our experiments the error incurred on our objective function by the approximate opinions was very small, with typically $\left|\widetilde{f} - f(X)\right| / f(X) < 10^{-8}$, where $\widetilde{f}$ is as in Corollary 4.

**Running time analysis of the optimization algorithms.** We start by comparing the running times of GDPM, BL-1, and BL-2, which use Algorithm 1 as a subroutine to compute approximate opinions $\widetilde{z}_X$, with an implementation that computes exact opinions $z_X$. We report our results in Table 4. While on TwitterSmall, the exact methods are still relatively fast, on TwitterLarge we observe that the algorithms with approximate opinions are faster by a factor of 300. In other words, running _all_ 150 iterations of GDPM with approximate opinions is faster than running _a single_ iteration with exact opinions.

Furthermore, in Fig. 6(b) we visualize the running time of a single iteration of GDPM, BL-1, and BL-2 and in Fig. 6(c) we plot the total time for 100 iterations of GDPM and 10 iterations of BL-1 and BL-2. The figures show that for all algorithms their running time grows linearly in the number of nodes. However, note that a single iteration of GDPM is faster than the baselines, particularly on large graphs. The reason is that for each row, BL-1 and BL-2 need to compute the topic indices $j$ and $j'$ which shall be favored and penalized (see Algorithm 3), which is costly; on the other hand, GDPM computes the gradient only once and the projection operation in GDPM for updating each row is highly efficient. This has the effect that as the size of the graphs increases, GDPM becomes more efficient than the baselines in terms of total running time, even though it performs 10 times more iterations.

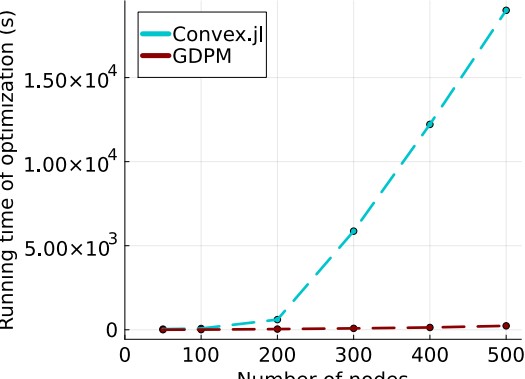

**Figure 7: Running time comparison of Convex.jl and GDPM on random graphs with synthetic opinions. We set GDPM to run 2,000 iterations and Convex.jl to run until the problem is solved. The objective value returned by both methods coincides with a precision of** $10^{-4}$**. We set** $C = 0.1$**,** $\theta = 0.1$ **for all experiments.**

**Comparison with black-box convex solver.** Figure 7 reports the comparison of the running time between GDPM and Convex.jl to solve Problem 3. Convex.jl is a popular black-box convex optimization tool written in Julia. It shows that GDPM outperforms the black-box solver in every experiment. Experiments are conducted on random graphs and the probability of creating an edge in the random graph is set to 0.5. We set the available memory resource for the experiments to be 102 GB. Running Convex.jl on a graph with nodes larger than 500 exceeds the memory constraint. So we only report the results of graphs with node sizes between 50 and 500. We generate synthetic opinions and user-to-topic matrix $\mathbf{X}$ and influence–topic matrix $\mathbf{Y}$ following the same method described in section B.1. The number of topics is set to 10 for all matrices. We set the learning rate $L = 100$ and run $2,000$ iterations for GDPM. In the Convex.jl experiment, the algorithm runs until the problem is solved or reaches the maximum iteration limit. In all experiments, Convex.jl and GDPM return the same objective value on the same setting after optimization with a precision of $10^{-4}$.

**Node degree changes after optimization.** In order to understand the node degree increase with intervention, we report node degree increase rate after optimization in TwitterSmall and TwitterLarge in Figure 8,. It shows that node degree increase rates are concentrated around 10% which is the same value of parameter $C$ we set. In Figure 8 (a)-(b), users are ranked in descending order by their corresponding summation of influence score among all topics, it shows that the user group with the highest influence scores has the largest standard deviation and higher mean. In Figure 8 (c)-(d), users are ranked in descending order by their node degree in the original graph, it shows that the mean of increase rate in groups with large node degrees is less than 10% (until group 12 in TwitterSmall and group 9 in TwitterLarge). One explanation is that we set the constraint to add 10% degree of the original graph. A small increase rate in nodes with large degrees will introduce a large absolute value into degrees, and a large increase rate in nodes with small degrees only introduces a small absolute value. And user groups with small degrees tend to have a larger standard deviation.

## C OMITTED PROOFS

### C.1 Preliminaries on linear algebra and optimization

We start by defining additional notation and recalling some basic facts from linear algebra and optimization.

We write $\lambda_i(\mathbf{X})$ to denote the $i$-th eigenvalue of $\mathbf{A}$. Similarly, $\sigma_i(\mathbf{X})$ denotes the $i$-th singular value of $\mathbf{X}$. We will sometimes also write $\lambda_{\min}(\mathbf{X})$, $\lambda_{\max}(\mathbf{X})$, $\sigma_{\min}(\mathbf{X})$ and $\sigma_{\max}(\mathbf{X})$ to denote the smallest and largest eigenvalues and singular values of $\mathbf{X}$, respectively.

Next, let us recall basic facts about matrix norms, where we let $\mathbf{X} \in \mathbb{R}^{m \times k}$, $\mathbf{Y} \in \mathbb{R}^{k \times n}$ and $\mathbf{v} \in \mathbb{R}^k$. Then we have that $\|\mathbf{X}\|_2 \leq \|\mathbf{X}\|_F$. Furthermore, it holds that $\|\mathbf{XY}\|_2 \leq \|\mathbf{X}\|_2 \cdot \|\mathbf{Y}\|_2$, as well as $\|\mathbf{XY}\|_F \leq \|\mathbf{X}\|_2 \cdot \|\mathbf{Y}\|_F$. We also have $\|\mathbf{Xv}\|_2 \leq \|\mathbf{X}\|_2 \cdot \|\mathbf{v}\|_2$. Furthermore, we denote the Frobenius scalar product by $\langle \mathbf{X}, \mathbf{Y} \rangle_F = \sum_{ij} \mathbf{X}_{ij} \mathbf{Y}_{ij}$.

If $\mathbf{X}, \mathbf{Y} \in \mathbb{R}^{n \times n}$ are invertible then observe that $\mathbf{X}^{-1} - \mathbf{Y}^{-1} = \mathbf{X}^{-1}(\mathbf{Y} - \mathbf{X})\mathbf{Y}^{-1}$, which based on the previous matrix inequalities implies that $\|\mathbf{X}^{-1} - \mathbf{Y}^{-1}\|_F \leq \|\mathbf{X}^{-1}\|_2 \cdot \|\mathbf{Y}^{-1}\|_2 \cdot \|\mathbf{X} - \mathbf{Y}\|_F$. Next, the Neumann series states that if $\|\mathbf{X}\|_2 < 1$ then $(\mathbf{I} - \mathbf{X})^{-1} = \sum_{i=0}^{\infty} \mathbf{X}^i$.

The *prox*-operator is given by

$$\text{prox}_f(x) = \arg\min_u \left\{ f(u) + \frac{1}{2} \|u - x\|^2 \right\}. \tag{5}$$

Given a convex set $Q$, we write $\delta_Q(x) \in \{0, \infty\}$ to denote its indicator function, i.e.,

$$\delta_Q(x) = \begin{cases} 0, & x \in Q, \\ \infty, & x \notin Q. \end{cases}$$

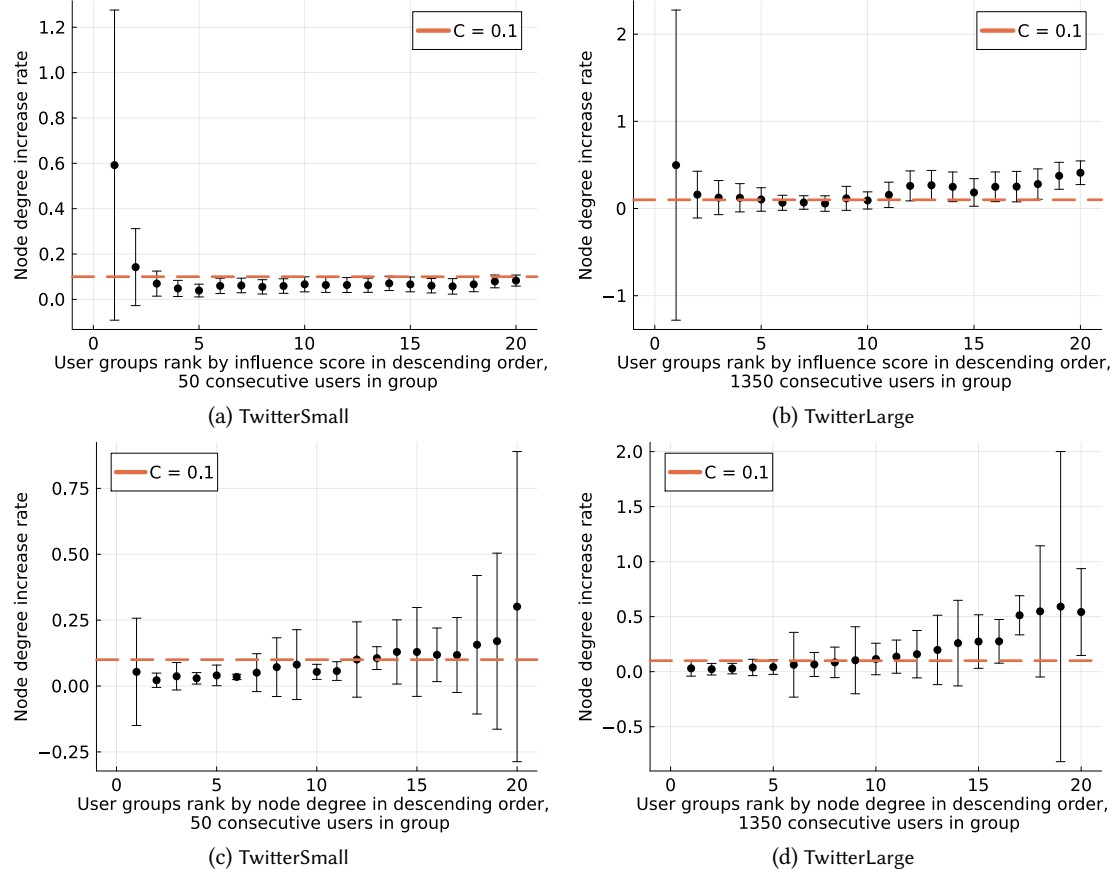

(a) TwitterSmall

(b) TwitterLarge

(c) TwitterSmall

(d) TwitterLarge

**Figure 8: Mean and standard deviation of node degree increase rate after optimization. The x-axis is user groups consisting of consecutive users in ranked lists in descending order, the y-axis is the node degree increase rate after adding the recommendation $A_X$ with optimal X. We set $C = 0.1$ in all experiments. In (a)-(b) users are ranked by influence scores among all topics. In (c)-(d) users are ranked by node degree in the original graph.**

## C.2 Useful facts

**Lemma 7.** *Let $M = D + L$, where $D \in \mathbb{R}^{n \times n}$ is a diagonal matrix with diagonal entries $D_{ii} \geq 1$ and $L$ is the Laplacian of a connected undirected weighted graph. Then all eigenvalues of $M$ are at least 1. Furthermore, for all $k \in \mathbb{N}$, the eigenvalues of $M^{-k}$ are most 1 and $\sum_{i=1}^{n} \lambda_i(M^{-k}) \leq n$.*

PROOF. Observe that all eigenvalues of $D$ are at least 1 since $D$ is diagonal and $D_{ii} \geq 1$ for all $i$. Furthermore, $L$ is positive semidefinite and thus all eigenvalues of $L$ are non-negative. Thus, Weyl's inequality implies that all eigenvalues of $D$ are at least 1.

The claim about the eigenvalues of $M^{-k}$ follows from the fact that the eigenvalues of $M^{-k}$ are given by $\lambda_1(M)^{-k}, \ldots, \lambda_n(M)^{-k}$ and the above argument implies that all of these numbers are at most 1. Summing over these eigenvalues gives their third claim of the lemma. □

**Lemma 8.** *Suppose $y, z \in [-1, 1]^n$ then $\|(y \odot y) - (z \odot z)\|_2 \leq 2 \|y - z\|_2$.*

PROOF. First, we note that for $a, b \in [-1, 1]$ it holds that $(a^2 - b^2)^2 \leq 4(a - b)^2$ since if $a = b$ the inequality clearly holds and if $a \neq b$ then

$$\frac{(a^2 - b^2)^2}{(a - b)^2} = \left(\frac{a^2 - b^2}{a - b}\right)^2 = \left(\frac{(a + b)(a - b)}{a - b}\right)^2 = (a + b)^2 \leq 4.$$

Using this inequality we get that

$$\|(y \odot y) - (z \odot z)\|_2^2 = \sum_{i=1}^{n} (y_i^2 - z_i^2)^2$$

$$\leq 4 \sum_{i=1}^{n} (y_i - z_i)^2$$

$$= 4 \|\mathbf{y} - \mathbf{z}\|_2^2.$$

We obtain the result by taking square roots. □

**Lemma 9.** *Suppose* $\mathbf{y}, \mathbf{z} \in [-1, 1]^n$ *then* $\left\| \mathbf{y} \cdot \mathbf{y}^\top - \mathbf{z} \cdot \mathbf{z}^\top \right\|_F \leq 2\sqrt{n} \|\mathbf{y} - \mathbf{z}\|_2$.

PROOF. We have that

$$\left\| \mathbf{y} \cdot \mathbf{y}^\top - \mathbf{z} \cdot \mathbf{z}^\top \right\|_F = \left\| (\mathbf{y} - \mathbf{z}) \cdot \mathbf{y}^\top + \mathbf{z} \cdot (\mathbf{y}^\top - \mathbf{z}^\top) \right\|_F$$
$$\leq \left\| (\mathbf{y} - \mathbf{z}) \cdot \mathbf{y}^\top \right\|_F + \left\| \mathbf{z} \cdot (\mathbf{y}^\top - \mathbf{z}^\top) \right\|_F$$
$$\leq (\|\mathbf{y}\|_2 + \|\mathbf{z}\|_2) \cdot \|\mathbf{y} - \mathbf{z}\|_2.$$

Since the entries of $\mathbf{y}$ and $\mathbf{z}$ are in $[-1, 1]$, we get that $\|\mathbf{y}\|_2 + \|\mathbf{z}\|_2 \leq 2\sqrt{n}$, which implies the lemma. □

The following lemma is a corollary of the Laplacian-solver technique by Koutis, Miller and Peng [22] and allows us to efficiently solve linear systems approximately.

**Lemma 10.** *Let* $\mathbf{D} \in \mathbb{R}^{n \times n}$ *be a diagonal matrix with entries* $\mathbf{D}_{ii} \geq 1$*, let* $\mathbf{L}$ *be the Laplacian of an undirected, connected, weighted graph, let* $\mathbf{b} \in \mathbb{R}^n$ *with* $\|\mathbf{b}\|_2 \leq \mathrm{poly}(n)$ *and let* $\epsilon > 0$*. Then there exists a function* $\mathrm{Solve}(\mathbf{D} + \mathbf{L}, \mathbf{b}, \epsilon)$ *that returns a vector* $\widetilde{\mathbf{x}} \in \mathbb{R}^n$ *such that* $\left\| \widetilde{\mathbf{x}} - (\mathbf{D} + \mathbf{L})^{-1}\mathbf{b} \right\|_2 \leq \epsilon$ *in time* $\widetilde{O}(m \log(1/\epsilon))$.

PROOF. For a symmetric matrix $\mathbf{A} \in \mathbb{R}^{n \times n}$, we write $\mathbf{A}^+$ to denote the Moore–Penrose pseudoinverse. We say that $\mathbf{A}$ is *diagonally dominant* if for all $i$, $\mathbf{A}_{ii} \geq \sum_{j \neq i} |\mathbf{A}_{ij}|$. For a vector $\mathbf{x} \in \mathbb{R}^n$ we set $\|\mathbf{x}\|_\mathbf{A} = \sqrt{\mathbf{x}^\top \mathbf{A} \mathbf{x}}$. We will use the following result by Koutis, Miller and Peng [22].

**Lemma 11** (Koutis, Miller, Peng [22]). *Let* $\mathbf{A} \in \mathbb{R}^{n \times n}$ *be a symmetric, diagonally dominant matrix with* $m$ *non-zero entries, let* $\mathbf{b} \in \mathbb{R}^n$ *and let* $\epsilon > 0$*. Then there exists a function* $\mathrm{Solve}(\mathbf{A}, \mathbf{b}, \epsilon)$*, which returns a vector* $\widetilde{\mathbf{x}} \in \mathbb{R}^{n \times n}$ *such that* $\left\| \widetilde{\mathbf{x}} - \mathbf{A}^+\mathbf{b} \right\|_\mathbf{A} \leq \epsilon \left\| \mathbf{A}^+\mathbf{b} \right\|_\mathbf{A}$ *in expected time* $\widetilde{O}(m \log(1/\epsilon))$.

We start with an observation about the norm $\|\mathbf{v}\|_\mathbf{A}$, where we use that diagonally dominant matrices are positive semidefinite:

$$\|\mathbf{v}\|_\mathbf{A} = \sqrt{\mathbf{v}^\top \mathbf{A} \mathbf{v}}$$
$$= \sqrt{\mathbf{v}^\top \mathbf{A}^{1/2} \mathbf{A}^{1/2} \mathbf{v}}$$
$$= \left\| \mathbf{A}^{1/2} \mathbf{v} \right\|_2$$
$$\geq \sigma_{\min}(\mathbf{A}^{1/2}) \|\mathbf{v}\|_2.$$

By rearranging terms we get that $\|\mathbf{v}\|_2 \leq \frac{1}{\sigma_{\min}(\mathbf{A}^{1/2})} \|\mathbf{v}\|_\mathbf{A}$ if $\sigma_{\min}(\mathbf{A}^{1/2}) > 0$.

We use the algorithm from Lemma 11 with $\mathbf{A} = \mathbf{D} + \mathbf{L}$, $\mathbf{b} = \mathbf{b}$ and $\epsilon' = \frac{\epsilon}{\|\mathbf{b}\|_2}$ to obtain a vector $\widetilde{\mathbf{x}}$. Note that since we assume that $\|\mathbf{b}\|_2 \leq \mathrm{poly}(n)$, we get that this algorithm runs in time $\widetilde{O}(m \log(1/\epsilon')) = \widetilde{O}(m \log(1/\epsilon))$, since this only adds additional $O(\log n)$-term, which is hidden in the $\widetilde{O}(\cdot)$-notation.

Now we observe that by Lemma 7, $\mathbf{D} + \mathbf{L}$ is positive definite with all eigenvalues at least 1. Thus $(\mathbf{D} + \mathbf{L})^+ = (\mathbf{D} + \mathbf{L})^{-1}$. Furthermore, the lemma implies that $\sigma_{\min}((\mathbf{D} + \mathbf{L})^{1/2}) = \lambda_{\min}(\mathbf{D} + \mathbf{L}) \geq 1$ and that all eigenvalues of $(\mathbf{D} + \mathbf{L})^{-1}$ are in the interval $(0, 1]$.

Now we use our previous result about $\|\mathbf{v}\|_\mathbf{A}$ to get that

$$\left\| \widetilde{\mathbf{x}} - (\mathbf{D} + \mathbf{L})^{-1}\mathbf{b} \right\|_2$$
$$\leq \frac{1}{\sigma_{\min}((\mathbf{D} + \mathbf{L})^{1/2})} \cdot \left\| \widetilde{\mathbf{x}} - (\mathbf{D} + \mathbf{L})^{-1}\mathbf{b} \right\|_{\mathbf{D}+\mathbf{L}}$$
$$\leq \frac{1}{\sigma_{\min}((\mathbf{D} + \mathbf{L})^{1/2})} \cdot \epsilon' \left\| (\mathbf{D} + \mathbf{L})^{-1}\mathbf{b} \right\|_{\mathbf{D}+\mathbf{L}}$$
$$= \frac{1}{\sigma_{\min}((\mathbf{D} + \mathbf{L})^{1/2})} \cdot \epsilon' \sqrt{\mathbf{b}^\top (\mathbf{D} + \mathbf{L})^{-1}\mathbf{b}}$$
$$\leq \frac{1}{\sigma_{\min}((\mathbf{D} + \mathbf{L})^{1/2})} \cdot \epsilon' \lambda_{\max}((\mathbf{D} + \mathbf{L})^{-1}) \|\mathbf{b}\|_2$$
$$\leq \epsilon' \cdot \|\mathbf{b}\|_2$$
$$= \epsilon.$$

□

## C.3 Proof of Lemma 1

First, recall that since $\mathbf{X}$ and $\mathbf{Y}$ are row-stochastic matrices with non-negative entries. Thus, we get that $\left\|\mathbf{XY} + \mathbf{Y}^T\mathbf{X}^T\right\|_{1,1} = \|\mathbf{XY}\|_{1,1} + \left\|\mathbf{Y}^T\mathbf{X}^T\right\|_{1,1}$.

Next, we have that

$$\|\mathbf{XY1}\|_{1,1} = \mathbf{1}^\top\mathbf{XY1}$$
$$= \mathbf{1}^\top\mathbf{X1}$$
$$= \mathbf{1}^\top\mathbf{1}$$
$$= n.$$

Similarly, since $\mathbf{Y}^\top$ and $\mathbf{X}^\top$ are column-stochastic, we get that

$$\left\|\mathbf{Y}^\top\mathbf{X}^\top\right\|_{1,1} = \mathbf{1}^\top\mathbf{Y}^\top\mathbf{X}^\top\mathbf{1}$$
$$= \mathbf{1}^\top\mathbf{X}^\top\mathbf{1}$$
$$= \mathbf{1}^\top\mathbf{1}$$
$$= n.$$

Summing over these two quantities proves the lemma.

## C.4 Proof of Proposition 3

Since this proof is rather involved, we first present a short proof sketch before giving the full details.

*C.4.1 Proof sketch.* Algorithm 1 is based on the observation that using the Woodbury matrix identity with $\mathbf{M} = \mathbf{I} + \mathbf{L} + \mathrm{diag}(\mathbf{A_X1})$, and $\mathbf{U}$ and $\mathbf{V}$ as before, we get that

$$\mathbf{z_X} = \mathbf{M}^{-1}\mathbf{s} + \frac{CW}{2n}\mathbf{M}^{-1}\mathbf{U}\left(\mathbf{I} - \frac{CW}{2n}\mathbf{VM}^{-1}\mathbf{U}\right)^{-1}\mathbf{VM}^{-1}\mathbf{s}.$$

Now Algorithm 1 basically computes this quantity from right to left. Our main insight here is that we can compute the quantities $\mathbf{M}^{-1}\mathbf{s}$ and $\mathbf{M}^{-1}\mathbf{U}$ using the Laplacian solver from Lemma 10. Here, we approximate $\mathbf{M}^{-1}\mathbf{U}$ column-by-column using the call Solve($\mathbf{M}, \mathbf{w}_j, \epsilon_\mathbf{R}$), where $\mathbf{w}_j$ is the $j$-th column of $\mathbf{U}$ and $\epsilon_\mathbf{R}$ is a suitable error parameter. The remaining matrix multiplications are efficient since $\mathbf{U}$ has only $2k$ columns and since $\mathbf{V}$ has only $2k$ rows.

To obtain our guarantees for the approximation error, we have to perform an intricate error analysis to ensure that errors do not compound too much. This is a challenge since we solve $\mathbf{I} - \frac{CW}{2n}\mathbf{VM}^{-1}\mathbf{U}$ only approximately but then we have to compute an inverse of this approximate quantity. In the proposition we use the assumptions that $\mathbf{VM}^{-1}\mathbf{U}$ exists and that $\left\|\mathbf{VM}^{-1}\mathbf{U}\right\|_2 \leq 0.99\frac{2n}{CW}$, to ensure that this can be done without obtaining too much error. In the proof we will also show that these assumptions imply that the inverse $\mathbf{S}^{-1}$ used in the algorithm exists.

*C.4.2 Formal proof.* We state the pseudocode of the algorithm in Algorithm 1.

Set $\mathbf{U} = \begin{pmatrix} \mathbf{X} & \mathbf{Y}^\top \end{pmatrix}$ and $\mathbf{V} = \begin{pmatrix} \mathbf{Y} \\ \mathbf{X}^\top \end{pmatrix}$. Now observe that $\frac{CW}{2n}\mathbf{UV} = \mathbf{A_X}$.

Recall that the Woodbury matrix identity states that

$$(\mathbf{M} + \mathbf{UCV})^{-1} = \mathbf{M}^{-1} - \mathbf{M}^{-1}\mathbf{U}(\mathbf{C}^{-1} + \mathbf{VM}^{-1}\mathbf{U})^{-1}\mathbf{VM}^{-1}.$$

Using the Woodbury matrix identity with $\mathbf{M} = \mathbf{I} + \mathbf{L} + \mathrm{diag}(\mathbf{A_X1})$, $\mathbf{C} = -\frac{CW}{2n}\mathbf{I}$, and $\mathbf{U}$ and $\mathbf{V}$ as before, we obtain that

$$\mathbf{z_X} = (\mathbf{I} + \mathbf{L} + \mathbf{L_X})^{-1}\mathbf{s}$$
$$= (\mathbf{I} + \mathbf{L} + \mathrm{diag}(\mathbf{A_X1}) - \mathbf{A_X})^{-1}\mathbf{s}$$
$$= \left(\mathbf{M} - \mathbf{U} \cdot \frac{CW}{2n}\mathbf{I} \cdot \mathbf{V}\right)^{-1}\mathbf{s}$$
$$= \mathbf{M}^{-1}\mathbf{s} - \mathbf{M}^{-1}\mathbf{U}\left(-\frac{2n}{CW}\mathbf{I} + \mathbf{VM}^{-1}\mathbf{U}\right)^{-1}\mathbf{VM}^{-1}\mathbf{s}$$
$$= \mathbf{M}^{-1}\mathbf{s} + \frac{CW}{2n}\mathbf{M}^{-1}\mathbf{U}\left(\mathbf{I} - \frac{CW}{2n}\mathbf{VM}^{-1}\mathbf{U}\right)^{-1}\mathbf{VM}^{-1}\mathbf{s},$$

where we use that $\mathbf{C}^{-1} = -\frac{2n}{CW}\mathbf{I}$.

We start by giving some intuition why Algorithm 1 computes an approximate version of $z_X$. We present the running time analysis and the formal error analysis below.

First, observe that $M$ is a symmetric, diagonally dominant matrix with $O(m)$ entries and diagonal entries $M_{ii} \geq 1$. Thus, whenever we wish to solve $M^{-1}v$, we can use the Laplacian solver from Lemma 10.

Next, note that $z_1$ is an approximation of $M^{-1}s$. Then $y_1$ becomes an approximation of $VM^{-1}s$. Now $R$ is an efficient approximation of $M^{-1}U$ by using Lemma 10. Hence, $S$ is an approximation of $I - \frac{CW}{2n}VM^{-1}U$ and $T$ approximates $(I - \frac{CW}{2n}VM^{-1}U)^{-1}$; we note that in the proof of Claim 13 we also point out why this inverse exists under the assumptions of the lemma. Continuing this approach, we obtain that $y_2$ approximates $\left(I - \frac{CW}{2n}VM^{-1}U\right)^{-1}VM^{-1}s$, $y_3$ approximates $U\left(I - \frac{CW}{2n}VM^{-1}U\right)^{-1}VM^{-1}s$, $z_2$ approximates $M^{-1}U^\top\left(I - \frac{CW}{2n}VM^{-1}U\right)^{-1}VM^{-1}s$, and hence $\widetilde{z}_X = z_1 + \frac{CW}{2n}z_2$ approximates $z_X$.

**Running time analysis.** Now let us consider the running time. Let us start by observing that to obtain our running time bounds we can only apply the Laplacian solvers on vectors $b$ with $\|b\|_2 \leq \text{poly}(n)$. Note that for Solve$(M, s, \epsilon_{z_1})$ and Solve$(M, w_i, \epsilon_R)$, $i = 1, \ldots, 2k$, this is clear since the entries in $s$ and $w_i$ are bounded by $[-1, 1]$ and $[0, 1]$, respectively. For Solve$(M, y_3, \epsilon_{z_2})$, we note that one can show that $\|y_3\| \leq \text{poly}(n)$ similar to the proofs of Claims 14 and 15 below.

First, note that diag$(A_X 1)$ can be computed in time $O(nk)$. Given diag$(A_X 1)$, we can compute $M$ in time $O(m)$ because it is the sum of $L$ and two diagonal matrices. Next, $z_1$ can be computed in time $\widetilde{O}(m \log(1/\epsilon_{z_1})) = \widetilde{O}(m \log(W/\epsilon))$ using the guarantee from Lemma 10. In the next step, $y_1 = Vz_1 \in \mathbb{R}^{2k}$ can be computed in time $O(nk)$ since $V$ is a $(2k) \times n$ matrix. Then by Lemma 10, $R$ can be computed in time $\widetilde{O}(mk \log(1/\epsilon_R)) = \widetilde{O}(mk \log(W/\epsilon))$ since we need $2k$ calls Solve$(M, w_j, \epsilon_R)$. As $I$ has only $O(k)$ non-zero entries and $V$ and $R$ are matrices of sizes $(2k) \times n$ and $n \times (2k)$, respectively, we can compute $S \in \mathbb{R}^{(2k) \times (2k)}$ in time $O(nk^2)$. Due to size of $S$, we can compute $T = S^{-1}$ in time $O(k^3)$ using Gaussian elimination and thus we obtain $y_2 \in \mathbb{R}^{2k}$ in time $O(k^2)$. Then we can compute $y_3$ in time $O(nk)$ since $U \in \mathbb{R}^{n \times (2k)}$ and we need time $\widetilde{O}(m \log(1/\epsilon_{z_2})) = \widetilde{O}(m \log(W/\epsilon))$ for the call to Solve$(M, y_3, \epsilon_{z_2})$ by Lemma 10. Summing over all terms above, we obtain our desired running time bound.

**Analysis of approximation error.** Let $z_2^*$ denote the error-free version of $z_2$, i.e.,

$$z_2^* = M^{-1}U\left(I - \frac{CW}{2n}VM^{-1}U\right)^{-1}VM^{-1}s.$$

Observe that the difference between $z_2$ and $z_2^*$ will be our main source of error when approximating $z_X$ with $\widetilde{z}_X$. Hence, next we write down $z_2$ to understand where we get inaccuracies compared to $z_2^*$.

First, we set $z_1^* = M^{-1}s$, i.e., it is the exact solution of $Mz_1^* = s$. Hence, we get that $z_1 = z_1^* + e_{z_1}$ where $e_{z_1}$ is the error vector introduced by the Laplacian solver. Thus, we get that $y_1 = Vz_1 = VM^{-1}s + Ve_{z_1}$. Next, let $E_T$ be the error matrix such that $T = \left(I - \frac{CW}{2n}VM^{-1}U\right)^{-1} + E_T$. Then we have that

$$y_2 = Ty_1 = \left(I - \frac{CW}{2n}VM^{-1}U\right)^{-1}VM^{-1}s + \left(\left(I - \frac{CW}{2n}VM^{-1}U\right)^{-1} + E_T\right)Ve_{z_1} + E_TVM^{-1}s.$$

Next,

$$y_3 = Uy_2 = U\left(I - \frac{CW}{2n}VM^{-1}U\right)^{-1}VM^{-1}s + U\left(\left(I - \frac{CW}{2n}VM^{-1}U\right)^{-1} + E_T\right)Ve_{z_1} + UE_TVM^{-1}s.$$

Finally, let $e_{z_2}$ be such that $z_2 = M^{-1}y_3 + e_{z_2}$. Then we get that

$$z_2 = M^{-1}y_3 + e_{z_2}$$

$$= M^{-1}U\left(I - \frac{CW}{2n}VM^{-1}U\right)^{-1}VM^{-1}s + M^{-1}U\left(\left(I - \frac{CW}{2n}VM^{-1}U\right)^{-1} + E_T\right)Ve_{z_1}$$

$$+ M^{-1}UE_TVM^{-1}s + e_{z_2}$$

$$= z_2^* + M^{-1}U\left(\left(I - \frac{CW}{2n}VM^{-1}U\right)^{-1} + E_T\right)Ve_{z_1} + M^{-1}UE_TVM^{-1}s + e_{z_2}.$$

The above implies that we return an approximation $\widetilde{z}_X$ such that

$$\|\widetilde{z}_X - z_X\|_2$$

$$= \left\| z_1 + \frac{CW}{2n} z_2 - z_1^* - \frac{CW}{2n} z_2^* \right\|_2$$

$$= \left\| e_{z_1} + \frac{CW}{2n} \left( M^{-1}U \left( \left( I - \frac{CW}{2n} VM^{-1}U \right)^{-1} + E_T \right) Ve_{z_1} + M^{-1}UE_TVM^{-1}s + e_{z_2} \right) \right\|_2 \tag{6}$$

$$\leq \|e_{z_1}\|_2 + \frac{CW}{2n} \left( \left\| M^{-1}U \left( \left( I - \frac{CW}{2n} VM^{-1}U \right)^{-1} + E_T \right) Ve_{z_1} \right\|_2 + \|M^{-1}UE_TVM^{-1}s\|_2 + \|e_{z_2}\|_2 \right).$$

Thus, for the remainder of the proof, we bound these four error terms.

**Claim 12.** $\left\| \left( I - \frac{CW}{2n} VM^{-1}U \right)^{-1} \right\|_2 \leq 100.$

PROOF. Using our assumption that $\|VM^{-1}U\|_2 \leq 0.99 \frac{2n}{CW}$, the Neumann series, triangle inequality and the geometric series, we obtain that

$$\left\| \left( I - \frac{CW}{2n} VM^{-1}U \right)^{-1} \right\|_2 = \left\| \sum_{i=0}^{\infty} \left( \frac{CW}{2n} VM^{-1}U \right)^i \right\|_2$$

$$\leq \sum_{i=0}^{\infty} \left( \frac{CW}{2n} \right)^i \|VM^{-1}U\|_2^i$$

$$\leq \sum_{i=0}^{\infty} 0.99^i$$

$$= \frac{1}{1 - 0.99}$$

$$= 100. \qquad \square$$

**Claim 13.** $\|E_T\|_2 \leq \min \left\{ 100, \frac{2n}{CW} \cdot \frac{\epsilon/4}{\|U\|_2 \cdot \|V\|_2 \cdot \|s\|_2} \right\}.$

PROOF. Let $E_R$ be the error matrix such that $R = M^{-1}U + E_R$ and recall that $T = \left( I - \frac{CW}{2n} VM^{-1}U \right)^{-1} + E_T$. Observe that $S = I - \frac{CW}{2n} VR = I - \frac{CW}{2n} V(M^{-1}U + E_R)$. Then we have that

$$\|E_T\|_2 = \left\| T - \left( I - \frac{CW}{2n} VM^{-1}U \right)^{-1} \right\|_2$$

$$= \left\| S^{-1} - \left( I + \frac{CW}{2n} VM^{-1}U \right)^{-1} \right\|_2$$

$$\leq \|S^{-1}\|_2 \cdot \left\| \left( I - \frac{CW}{2n} VM^{-1}U \right)^{-1} \right\|_2 \cdot \left\| S - \left( I - \frac{CW}{2n} VM^{-1}U \right) \right\|_2$$

$$= \|S^{-1}\|_2 \cdot \left\| \left( I - \frac{CW}{2n} VM^{-1}U \right)^{-1} \right\|_2 \cdot \left\| \frac{CW}{2n} VE_R \right\|_2$$

$$\leq \frac{CW}{2n} \|S^{-1}\|_2 \cdot \left\| \left( I - \frac{CW}{2n} VM^{-1}U \right)^{-1} \right\|_2 \cdot \|V\|_2 \cdot \|E_R\|_2.$$

Next, let $e_{w_j}$ denote the error in the $j$-th column of $R$, i.e., $e_{w_j}$ is the $j$-th column of $E_R$. Observe that by Lemma 10 and our choice of $\epsilon_R = \frac{1}{2k} \min \left\{ 0.009 \frac{2n}{CW \cdot \|V\|_F}, \frac{2n}{10^5 \cdot CW \cdot \|V\|_2} \cdot \min \left\{ 100, \frac{2n}{CW} \cdot \frac{\epsilon/4}{\|U\|_2 \cdot \|V\|_2 \cdot \|s\|_2} \right\} \right\}$, we get that $\|e_{w_j}\|_2 \leq \epsilon_R$ for all $j$. Then we get that

$$\|E_R\|_2 \leq \|E_R\|_F$$

$$= \sqrt{\sum_j \sum_i (E_R)_{ij}^2}$$

$$= \sqrt{\sum_j \left\| \mathbf{e}_{\mathbf{w}_j} \right\|_2^2}$$

$$\le \sum_j \left\| \mathbf{e}_{\mathbf{w}_j} \right\|_2$$

$$\le 2k \cdot \epsilon_{\mathbf{R}}$$

$$\le \min \left\{ 0.009 \frac{2n}{CW \cdot \|\mathbf{V}\|_F}, \frac{2n}{10^5 \cdot CW \cdot \|\mathbf{V}\|_2} \cdot \min \left\{ 100, \frac{2n}{CW} \cdot \frac{\epsilon/4}{\|\mathbf{U}\|_2 \cdot \|\mathbf{V}\|_2 \cdot \|\mathbf{s}\|_2} \right\} \right\},$$

where the fourth step holds since $\|\mathbf{v}\|_2 \le \|\mathbf{v}\|_1$ for any vector $\mathbf{v} \in \mathbb{R}^n$.

Thus, similar to the proof of Claim 12 we obtain that

$$\left\| \mathbf{S}^{-1} \right\|_2 = \left\| \left( \mathbf{I} - \frac{CW}{2n} \mathbf{V} (\mathbf{M}^{-1} \mathbf{U} + \mathbf{E}_{\mathbf{R}}) \right)^{-1} \right\|_2$$

$$\le \sum_{i=0}^{\infty} \left( \frac{CW}{2n} \right)^i \left\| \mathbf{V} \mathbf{M}^{-1} \mathbf{U} + \mathbf{V} \mathbf{E}_{\mathbf{R}} \right\|_2^i$$

$$\le \sum_{i=0}^{\infty} \left( \frac{CW}{2n} \right)^i \left( \left\| \mathbf{V} \mathbf{M}^{-1} \mathbf{U} \right\|_2 + \|\mathbf{V}\|_F \|\mathbf{E}_{\mathbf{R}}\|_F \right)^i$$

$$\le \sum_{i=0}^{\infty} \left( \frac{CW}{2n} \right)^i \left( 0.999 \frac{2n}{CW} \right)^i$$

$$= \sum_{i=0}^{\infty} 0.999^i$$

$$= 10^3.$$

Note that this implies that $\mathbf{S}^{-1}$ exists.

Now combining the above results with Claim 12, we get that

$$\|\mathbf{E}_{\mathbf{T}}\|_2 \le \frac{CW}{2n} \left\| \mathbf{S}^{-1} \right\|_2 \cdot \left\| \left( \mathbf{I} - \frac{CW}{2n} \mathbf{V} \mathbf{M}^{-1} \mathbf{U} \right)^{-1} \right\|_2 \cdot \|\mathbf{V}\|_2 \cdot \|\mathbf{E}_{\mathbf{R}}\|_2$$

$$\le \frac{CW}{2n} \cdot 10^3 \cdot 10^2 \cdot \|\mathbf{V}\|_2 \cdot \frac{2n}{10^5 \cdot CW \cdot \|\mathbf{V}\|_2} \cdot \min \left\{ 100, \frac{2n}{CW} \cdot \frac{\epsilon/4}{\|\mathbf{U}\|_2 \cdot \|\mathbf{V}\|_2 \cdot \|\mathbf{s}\|_2} \right\}$$

$$\le \min \left\{ 100, \frac{2n}{CW} \cdot \frac{\epsilon/4}{\|\mathbf{U}\|_2 \cdot \|\mathbf{V}\|_2 \cdot \|\mathbf{s}\|_2} \right\}. \qquad \square$$

**Claim 14.** $\frac{CW}{2n} \left\| \mathbf{M}^{-1} \mathbf{U} \mathbf{E}_{\mathbf{T}} \mathbf{V} \mathbf{M}^{-1} \mathbf{s} \right\|_2 \le \epsilon/4.$

PROOF. First, observe that $\left\| \mathbf{M}^{-1} \right\|_2 \le 1$ by Lemma 7. Now using Claim 13 we get that

$$\frac{CW}{2n} \left\| \mathbf{M}^{-1} \mathbf{U} \mathbf{E}_{\mathbf{T}} \mathbf{V} \mathbf{M}^{-1} \mathbf{s} \right\|_2 \le \frac{CW}{2n} \cdot \left\| \mathbf{M}^{-1} \right\|_2^2 \cdot \|\mathbf{U}\|_2 \cdot \|\mathbf{V}\|_2 \cdot \|\mathbf{s}\|_2 \cdot \|\mathbf{E}_{\mathbf{T}}\|_2$$

$$\le \frac{CW}{2n} \cdot \|\mathbf{U}\|_2 \cdot \|\mathbf{V}\|_2 \cdot \|\mathbf{s}\|_2 \cdot \|\mathbf{E}_{\mathbf{T}}\|_2$$

$$\le \epsilon/4. \qquad \square$$

**Claim 15.** $\frac{CW}{2n} \left\| \mathbf{M}^{-1} \mathbf{U} \left( \left( \mathbf{I} - \frac{CW}{2n} \mathbf{V} \mathbf{M}^{-1} \mathbf{U} \right)^{-1} + \mathbf{E}_{\mathbf{T}} \right) \mathbf{V} \mathbf{e}_{\mathbf{z}_1} \right\|_2 \le \epsilon/4.$

PROOF. First, observe that by Claims 12 and 13 and the triangle inequality,

$$\left\| \left( \mathbf{I} - \frac{CW}{2n} \mathbf{V} \mathbf{M}^{-1} \mathbf{U} \right)^{-1} + \mathbf{E}_{\mathbf{T}} \right\|_2 \le \left\| \left( \mathbf{I} - \frac{CW}{2n} \mathbf{V} \mathbf{M}^{-1} \mathbf{U} \right)^{-1} \right\|_2 + \|\mathbf{E}_{\mathbf{T}}\|_2 \le 200.$$

Using the inequality from above and Lemma 7 and our choice of $\epsilon_{\mathbf{z}_1} = \frac{\epsilon}{4} \cdot \min \left\{ 1, \frac{2n}{200 \cdot CW \cdot \|\mathbf{U}\|_2 \cdot \|\mathbf{V}\|_2} \right\}$, we obtain that

$$\frac{CW}{2n} \left\| \mathbf{M}^{-1} \mathbf{U} \left( \left( \mathbf{I} - \frac{CW}{2n} \mathbf{V} \mathbf{M}^{-1} \mathbf{U} \right)^{-1} + \mathbf{E}_{\mathbf{T}} \right) \mathbf{V} \mathbf{e}_{\mathbf{z}_1} \right\|_2$$

$$\leq \frac{CW}{2n} \cdot \left\| \mathbf{M}^{-1} \right\|_2 \cdot \left\| \mathbf{U} \right\|_2 \cdot \left\| \left( \mathbf{I} - \frac{CW}{2n} \mathbf{V} \mathbf{M}^{-1} \mathbf{U} \right)^{-1} + \mathbf{E_T} \right\|_2 \cdot \left\| \mathbf{V} \right\|_2 \cdot \left\| \mathbf{e_{z_1}} \right\|_2$$

$$\leq \frac{CW}{2n} \cdot \left\| \mathbf{U} \right\|_2 \cdot 200 \cdot \left\| \mathbf{V} \right\|_2 \cdot \left\| \mathbf{e_{z_1}} \right\|_2$$

$$\leq \epsilon/4. \qquad\qquad\qquad \square$$

Now continuing Eq. (6), using Claims 15 and 14, and our choices $\epsilon_{\mathbf{z_1}} = \frac{\epsilon}{4} \cdot \min\left\{ 1, \frac{2n}{200 \cdot CW \cdot \|\mathbf{U}\|_2 \cdot \|\mathbf{V}\|_2} \right\}$ and $\epsilon_{\mathbf{z_2}} = \frac{2n}{CW} \cdot \frac{\epsilon}{4}$, we get that

$$\left\| \widetilde{\mathbf{z}}_{\mathbf{X}} - \mathbf{z_X} \right\|_2$$

$$\leq \left\| \mathbf{e_{z_1}} \right\|_2 + \frac{CW}{2n} \left( \left\| \mathbf{M}^{-1} \mathbf{U} \left( \left( \mathbf{I} - \frac{CW}{2n} \mathbf{V} \mathbf{M}^{-1} \mathbf{U} \right)^{-1} + \mathbf{E_T} \right) \mathbf{V} \mathbf{e_{z_1}} \right\|_2 + \left\| \mathbf{M}^{-1} \mathbf{U} \mathbf{E_T} \mathbf{V} \mathbf{M}^{-1} \mathbf{s} \right\|_2 + \left\| \mathbf{e_{z_2}} \right\|_2 \right)$$

$$\leq \epsilon/4 + \epsilon/4 + \epsilon/4 + \epsilon/4$$

$$= \epsilon.$$

## C.5 Proof of Corollary 4

We compute $\widetilde{\mathbf{z}}_{\mathbf{X}}$ using Proposition 3 with $\epsilon' = \frac{\epsilon}{\sqrt{n}}$. Then we set $\widetilde{f} = \mathbf{s}^\top \widetilde{\mathbf{z}}_{\mathbf{X}}$. Using the Cauchy–Schwarz inequality, we obtain that

$$\left| \widetilde{f} - f(\mathbf{X}) \right| = \left| \mathbf{s}^\top \widetilde{\mathbf{z}}_{\mathbf{X}} - \mathbf{s}^\top \mathbf{z_X} \right|$$

$$= \left| \mathbf{s}^\top (\widetilde{\mathbf{z}}_{\mathbf{X}} - \mathbf{z_X}) \right|$$

$$\leq \left\| \mathbf{s} \right\|_2 \cdot \left\| \widetilde{\mathbf{z}}_{\mathbf{X}} - \mathbf{z_X} \right\|_2$$

$$\leq \sqrt{n} \cdot \frac{\epsilon}{\sqrt{n}}$$

$$= \epsilon,$$

where we use that all entries in $\mathbf{s}$ are in the interval $[-1, 1]$.

## C.6 Proof of Proposition 5

### C.6.1 Derivation of the gradient.

*Matrixcalculus.* We use matrixcalculus.org [24, 25] to obtain the gradient. We set $F = \mathbf{I} + \mathbf{L}$, $c = \frac{CW}{2n}$, $v = \mathbf{1}$ and use the input s'*inv(F + diag (c*(X*Y + Y'*X')*v) - c*(X*Y + Y'*X')) * s.

We obtain:

$$\frac{\partial}{\partial X} \left( s^\top \cdot \mathrm{inv}(F + \mathrm{diag}(c \cdot (X \cdot Y + Y^\top \cdot X^\top) \cdot v) - c \cdot (X \cdot Y + Y^\top \cdot X^\top)) \cdot s \right) =$$

$$-(c \cdot t_5 \odot t_7 \cdot (Y \cdot v)^\top + c \cdot v \cdot ((t_8 \odot t_6) \cdot Y^\top) - (c \cdot t_5 \cdot (t_6 \cdot Y^\top) + c \cdot t_7 \cdot (t_8 \cdot Y^\top)))$$

where

- $T_0 = X \cdot Y$
- $T_1 = T_0^\top + T_0$
- $T_2 = T_0^\top + T_0$
- $T_3 = \mathrm{inv}(F + c \cdot \mathrm{diag}(T_2 \cdot v) - c \cdot T_2)$
- $T_4 = \mathrm{inv}(F^\top + c \cdot \mathrm{diag}(v^\top \cdot T_1) - c \cdot T_1)$
- $t_5 = T_4 \cdot s$
- $t_6 = s^\top \cdot T_4$
- $t_7 = T_3 \cdot s$
- $t_8 = s^\top \cdot T_3$

and

- $F$ is a symmetric matrix
- $X$ is a matrix
- $Y$ is a matrix
- $c$ is a scalar
- $s$ is a vector
- $v$ is a vector

*Simplification 1.* We note above that $T_1 = T_2$, hence we replace ever occurence of $T_2$ with $T_1$. Furthermore, in our setting we have that $F$ and $T_1$ are symmetric $n \times n$ matrices, which implies that $T_3 = T_4$; hence, we replace every occurence of $T_4$ with $T_3$.

Then we get:

$$\frac{\partial}{\partial X} \left( s^\top \cdot \text{inv}(F + \text{diag}(c \cdot (X \cdot Y + Y^\top \cdot X^\top)) \cdot v) - c \cdot (X \cdot Y + Y^\top \cdot X^\top)) \cdot s \right) =$$
$$-(c \cdot t_5 \odot t_7 \cdot (Y \cdot v)^\top + c \cdot v \cdot ((t_8 \odot t_6) \cdot Y^\top) - (c \cdot t_5 \cdot (t_6 \cdot Y^\top) + c \cdot t_7 \cdot (t_8 \cdot Y^\top)))$$

where
- $T_0 = X \cdot Y$
- $T_1 = T_0^\top + T_0$
- $T_3 = \text{inv}(F + c \cdot \text{diag}(T_1 \cdot v) - c \cdot T_1)$
- $t_5 = T_3 \cdot s$
- $t_6 = s^\top \cdot T_3$
- $t_7 = T_3 \cdot s$
- $t_8 = s^\top \cdot T_3$

*Simplification 2.* Next, observe that $t_5 = t_7$ and $t_6 = t_8$. Hence, we only use $t_5$ and $t_6$.

Then we get:

$$\frac{\partial}{\partial X} \left( s^\top \cdot \text{inv}(F + \text{diag}(c \cdot (X \cdot Y + Y^\top \cdot X^\top) \cdot v) - c \cdot (X \cdot Y + Y^\top \cdot X^\top)) \cdot s \right) =$$
$$-(c \cdot t_5 \odot t_5 \cdot (Y \cdot v)^\top + c \cdot v \cdot ((t_6 \odot t_6) \cdot Y^\top) - (c \cdot t_5 \cdot (t_6 \cdot Y^\top) + c \cdot t_5 \cdot (t_6 \cdot Y^\top)))$$

where
- $T_0 = X \cdot Y$
- $T_1 = T_0^\top + T_0$
- $T_3 = \text{inv}(F + c \cdot \text{diag}(T_1 \cdot v) - c \cdot T_1)$
- $t_5 = T_3 \cdot s$
- $t_6 = s^\top \cdot T_3$

*Simplification 3.* Next, we remove the leading minus sign by multiplying it inside. We also observe that $t_5 = t_6^\top$ since $T_3$ is symmetric. Hence, we only use $t_5$ and $t_5^\top$.

Then we get:

$$\frac{\partial}{\partial X} \left( s^\top \cdot \text{inv}(F + \text{diag}(c \cdot (X \cdot Y + Y^\top \cdot X^\top)) \cdot v) - c \cdot (X \cdot Y + Y^\top \cdot X^\top)) \cdot s \right) =$$
$$-c \cdot t_5 \odot t_5 \cdot (Y \cdot v)^\top - c \cdot v \cdot ((t_5^\top \odot t_5^\top) \cdot Y^\top) + c \cdot t_5 \cdot (t_5^\top \cdot Y^\top) + c \cdot t_5 \cdot (t_5^\top \cdot Y^\top)$$

where
- $T_0 = X \cdot Y$
- $T_1 = T_0^\top + T_0$
- $T_3 = \text{inv}(F + c \cdot \text{diag}(T_1 \cdot v) - c \cdot T_1)$
- $t_5 = T_3 \cdot s$

*Simplification 4.* Next, we first observe that the final two terms above are the same. Also, recall that we set $v = \mathbf{1}$ and that $Y$ is a row-stochastic $k \times n$ matrix. Hence, we get that $Yv = Y\mathbf{1} = \mathbf{1}_k$, where $\mathbf{1}_k \in \mathbb{R}^k$ is a row-vector in which all entries are set to 1. We also substitute our notation from above and observe that $T_3 = \text{inv}(F + c \cdot \text{diag}(T_1 \cdot v) - c \cdot T_1) = (\mathbf{I} + \mathbf{L} + \mathbf{L}_\mathbf{X})^{-1}$.

Then we get:

$$\frac{\partial}{\partial X} \left( s^\top \cdot \text{inv}(F + \text{diag}(c \cdot (X \cdot Y + Y^\top \cdot X^\top)) \cdot v) - c \cdot (X \cdot Y + Y^\top \cdot X^\top)) \cdot s \right) =$$
$$-c \cdot t_5 \odot t_5 \cdot \mathbf{1}_k^\top - c \cdot \mathbf{1} \cdot ((t_5^\top \odot t_5^\top) \cdot Y^\top) + 2c \cdot t_5 \cdot (t_5^\top \cdot Y^\top)$$

where
- $t_5 = (\mathbf{I} + \mathbf{L} + \mathbf{L}_\mathbf{X})^{-1} \cdot s$

*Simplification 5.* Observing that $t_5 = \mathbf{z}_\mathbf{X}$, we obtain at our final gradient.

Then we get:

$$\nabla_\mathbf{X} \left( s^\top (\mathbf{I} + \mathbf{L} + \mathbf{L}_\mathbf{X})^{-1} s \right) = \frac{CW}{2n} (2 \cdot \mathbf{z}_\mathbf{X} \cdot (\mathbf{z}_\mathbf{X}^\top \cdot \mathbf{Y}^\top) - \mathbf{z}_\mathbf{X} \odot \mathbf{z}_\mathbf{X} \cdot \mathbf{1}_k^\top - \mathbf{1}((\mathbf{z}_\mathbf{X}^\top \odot \mathbf{z}_\mathbf{X}^\top) \cdot \mathbf{Y}^\top))$$

*C.6.2 The gradient is Lipschitz.* We need to show that $\|\nabla_{\mathbf{X}} f(\mathbf{X}_1) - \nabla_{\mathbf{X}} f(\mathbf{X}_2)\|_F \leq L \|\mathbf{X}_1 - \mathbf{X}_2\|_F$ for all $\mathbf{X}_1, \mathbf{X}_2 \in Q$.

Using the previously derived gradient and the triangle inequality, we get that

$$
\begin{aligned}
\|\nabla_{\mathbf{X}} f(\mathbf{X}_1) - \nabla_{\mathbf{X}} f(\mathbf{X}_2)\|_F \leq \frac{CW}{2n} \big[ &\left\| 2 \cdot \mathbf{z}_{\mathbf{X}_1} \cdot (\mathbf{z}_{\mathbf{X}_1}^\top \cdot \mathbf{Y}^\top) - 2 \cdot \mathbf{z}_{\mathbf{X}_2} \cdot (\mathbf{z}_{\mathbf{X}_2}^\top \cdot \mathbf{Y}^\top) \right\|_F \\
&+ \left\| \mathbf{z}_{\mathbf{X}_1} \odot \mathbf{z}_{\mathbf{X}_1} \cdot \mathbf{1}_k^\top - \mathbf{z}_{\mathbf{X}_2} \odot \mathbf{z}_{\mathbf{X}_2} \cdot \mathbf{1}_k^\top \right\|_F \\
&+ \left\| \mathbf{1}_n ((\mathbf{z}_{\mathbf{X}_1}^\top \odot \mathbf{z}_{\mathbf{X}_1}^\top) \cdot \mathbf{Y}^\top) - \mathbf{1}_n ((\mathbf{z}_{\mathbf{X}_2}^\top \odot \mathbf{z}_{\mathbf{X}_2}^\top) \cdot \mathbf{Y}^\top)) \right\|_F \big] \\
= \frac{CW}{2n} \big[ &\left\| 2 \cdot (\mathbf{z}_{\mathbf{X}_1} \cdot \mathbf{z}_{\mathbf{X}_1}^\top - \mathbf{z}_{\mathbf{X}_2} \cdot \mathbf{z}_{\mathbf{X}_2}^\top) \mathbf{Y}^\top) \right\|_F \\
&+ \left\| (\mathbf{z}_{\mathbf{X}_1} \odot \mathbf{z}_{\mathbf{X}_1} - \mathbf{z}_{\mathbf{X}_2} \odot \mathbf{z}_{\mathbf{X}_2}) \mathbf{1}_k^\top) \right\|_F \\
&+ \left\| \mathbf{1}_n (\mathbf{z}_{\mathbf{X}_1}^\top \odot \mathbf{z}_{\mathbf{X}_1}^\top - \mathbf{z}_{\mathbf{X}_2}^\top \odot \mathbf{z}_{\mathbf{X}_2}^\top) \cdot \mathbf{Y}^\top \right\|_F \big]
\end{aligned}
$$

Next, we will now bound each of these terms invidually.

We start by making a crucial observation about the difference of $\mathbf{z}_{\mathbf{X}_1}$ and $\mathbf{z}_{\mathbf{X}_2}$, where we use Lemma 7 in the final step:

$$
\begin{aligned}
&\left\| \mathbf{z}_{\mathbf{X}_1} - \mathbf{z}_{\mathbf{X}_2} \right\|_F \\
&= \left\| (\mathbf{I} + \mathbf{L} + \mathbf{L}_{\mathbf{X}_1})^{-1} \mathbf{s} - (\mathbf{I} + \mathbf{L} + \mathbf{L}_{\mathbf{X}_2})^{-1} \mathbf{s} \right\|_F \\
&\leq \|\mathbf{s}\|_2 \cdot \left\| (\mathbf{I} + \mathbf{L} + \mathbf{L}_{\mathbf{X}_1})^{-1} - (\mathbf{I} + \mathbf{L} + \mathbf{L}_{\mathbf{X}_2})^{-1} \right\|_F \\
&\leq \|\mathbf{s}\|_2 \cdot \left\| (\mathbf{I} + \mathbf{L} + \mathbf{L}_{\mathbf{X}_1})^{-1} \right\|_2 \cdot \left\| (\mathbf{I} + \mathbf{L} + \mathbf{L}_{\mathbf{X}_2})^{-1} \right\|_2 \cdot \left\| (\mathbf{I} + \mathbf{L} + \mathbf{L}_{\mathbf{X}_1}) - (\mathbf{I} + \mathbf{L} + \mathbf{L}_{\mathbf{X}_2}) \right\|_F \\
&= 2 \|\mathbf{s}\|_2 \cdot \left\| (\mathbf{I} + \mathbf{L} + \mathbf{L}_{\mathbf{X}_1})^{-1} \right\|_2 \cdot \left\| (\mathbf{I} + \mathbf{L} + \mathbf{L}_{\mathbf{X}_2})^{-1} \right\|_2 \cdot \left\| (\mathbf{X}_1 - \mathbf{X}_2) \mathbf{Y} \right\|_F \\
&\leq 2 \|\mathbf{s}\|_2 \cdot \|\mathbf{Y}\|_2 \cdot \|\mathbf{X}_1 - \mathbf{X}_2\|_F .
\end{aligned}
$$

Next, observe that for all $\mathbf{X}$ we have that $\|\mathbf{z}_{\mathbf{X}}\|_2 \leq \sqrt{n}$ since $\|\mathbf{z}_{\mathbf{X}}\|_2 \leq \left\| (\mathbf{I} + \mathbf{L} + \mathbf{L}_{\mathbf{X}_1})^{-1} \right\|_2 \cdot \|\mathbf{s}\|_2 \leq \|\mathbf{s}\|_2 \leq \sqrt{n}$, where we use Lemma 7 and the fact that the entries in $\mathbf{s}$ are in $[-1, 1]$. In particular, this implies that $\left\| \mathbf{z}_{\mathbf{X}_1} \right\|_2 + \left\| \mathbf{z}_{\mathbf{X}_2} \right\|_2 \leq 2\sqrt{n}$.

Now using Lemma 9 together with $\left\| \mathbf{z}_{\mathbf{X}_1} \right\|_2 + \left\| \mathbf{z}_{\mathbf{X}_2} \right\|_2 \leq 2\sqrt{n}$ and our inequalty from above, we obtain that

$$
\begin{aligned}
&\left\| 2 \cdot (\mathbf{z}_{\mathbf{X}_1} \cdot \mathbf{z}_{\mathbf{X}_1}^\top - \mathbf{z}_{\mathbf{X}_2} \cdot \mathbf{z}_{\mathbf{X}_2}^\top) \mathbf{Y}^\top \right\|_F \\
&\leq 2 \|\mathbf{Y}\|_2 \cdot \left\| \mathbf{z}_{\mathbf{X}_1} \cdot \mathbf{z}_{\mathbf{X}_1}^\top - \mathbf{z}_{\mathbf{X}_2} \cdot \mathbf{z}_{\mathbf{X}_2}^\top \right\|_F \\
&\leq 4\sqrt{n} \cdot \|\mathbf{Y}\|_2 \cdot \left\| \mathbf{z}_{\mathbf{X}_1} - \mathbf{z}_{\mathbf{X}_2} \right\|_F \\
&\leq 8\sqrt{n} \cdot \|\mathbf{s}\|_2 \cdot \|\mathbf{Y}\|_2^2 \cdot \|\mathbf{X}_1 - \mathbf{X}_2\|_F .
\end{aligned}
$$

Next, we show a fact about $\mathbf{z}_{\mathbf{X}_1} \odot \mathbf{z}_{\mathbf{X}_1} - \mathbf{z}_{\mathbf{X}_2} \odot \mathbf{z}_{\mathbf{X}_2}$. Using Lemma 8 and our inequality from above, we get that

$$
\begin{aligned}
&\left\| \mathbf{z}_{\mathbf{X}_1} \odot \mathbf{z}_{\mathbf{X}_1} - \mathbf{z}_{\mathbf{X}_2} \odot \mathbf{z}_{\mathbf{X}_2} \right\|_2 \\
&\leq 2 \left\| \mathbf{z}_{\mathbf{X}_1} - \mathbf{z}_{\mathbf{X}_2} \right\|_2 \\
&\leq 4 \|\mathbf{s}\|_2 \cdot \|\mathbf{Y}\|_2 \cdot \|\mathbf{X}_1 - \mathbf{X}_2\|_F .
\end{aligned}
$$

Using the inequality from above, we get that

$$
\begin{aligned}
&\left\| (\mathbf{z}_{\mathbf{X}_1} \odot \mathbf{z}_{\mathbf{X}_1} - \mathbf{z}_{\mathbf{X}_2} \odot \mathbf{z}_{\mathbf{X}_2}) \mathbf{1}_k^\top \right\|_F \\
&= \sqrt{k} \cdot \left\| \mathbf{z}_{\mathbf{X}_1} \odot \mathbf{z}_{\mathbf{X}_1} - \mathbf{z}_{\mathbf{X}_2} \odot \mathbf{z}_{\mathbf{X}_2} \right\|_2 \\
&\leq 4\sqrt{k} \cdot \|\mathbf{s}\|_2 \cdot \|\mathbf{Y}\|_2 \cdot \|\mathbf{X}_1 - \mathbf{X}_2\|_F .
\end{aligned}
$$

Furthermore, we again use the inequality from above to obtain that

$$
\begin{aligned}
&\left\| \mathbf{1}_n (\mathbf{z}_{\mathbf{X}_1}^\top \odot \mathbf{z}_{\mathbf{X}_1}^\top - \mathbf{z}_{\mathbf{X}_2}^\top \odot \mathbf{z}_{\mathbf{X}_2}^\top) \cdot \mathbf{Y}^\top \right\|_F \\
&\leq \|\mathbf{Y}\|_2 \cdot \left\| \mathbf{1}_n (\mathbf{z}_{\mathbf{X}_1}^\top \odot \mathbf{z}_{\mathbf{X}_1}^\top - \mathbf{z}_{\mathbf{X}_2}^\top \odot \mathbf{z}_{\mathbf{X}_2}^\top) \right\|_F \\
&= \|\mathbf{Y}\|_2 \cdot \sqrt{n} \cdot \left\| \mathbf{z}_{\mathbf{X}_1}^\top \odot \mathbf{z}_{\mathbf{X}_1}^\top - \mathbf{z}_{\mathbf{X}_2}^\top \odot \mathbf{z}_{\mathbf{X}_2}^\top \right\|_2 \\
&\leq 4\sqrt{n} \cdot \|\mathbf{s}\|_2 \cdot \|\mathbf{Y}\|_2^2 \cdot \|\mathbf{X}_1 - \mathbf{X}_2\|_F .
\end{aligned}
$$

By combining the results from above and using our assumption that $k \leq n$, we get that the gradient is Lipschitz with $L = \frac{8CW}{\sqrt{n}} \cdot \|\mathbf{s}\|_2 \cdot \|\mathbf{Y}\|_2^2$.

*C.6.3 Approximate gradient.* We use Proposition 3 with $\epsilon' = \frac{1}{8} \cdot \frac{\min\{\epsilon, \sqrt{\epsilon}\} \cdot \sqrt{n}}{(1+CW)(1+\|\mathbf{Y}\|_F)}$ to obtain $\widetilde{\mathbf{z}}_\mathbf{X}$. Note that in the numerator we use $\min\{\epsilon, \sqrt{\epsilon}\}$ since it might be that $\sqrt{\epsilon} > \epsilon$ for $\epsilon < 1$; in denominator we use the terms $1 + CW$ and $1 + \|\mathbf{Y}\|_F$ because it is possible that $CW < 1$ and also $\|\mathbf{Y}\|_F < 1$. Observe that with this choice of $\epsilon'$, Proposition 3 guarantees a running time of $\widetilde{O}((mk + nk^2 + k^3)\log(W/\epsilon))$.

Now we consider the approximate gradient

$$\widetilde{\nabla}_\mathbf{X} f(\mathbf{X}, C) = \frac{CW}{2n}(2 \cdot \widetilde{\mathbf{z}}_{\mathbf{X}(T)} \cdot \widetilde{\mathbf{z}}_{\mathbf{X}(T)}^\top \cdot \mathbf{Y}^\top - (\widetilde{\mathbf{z}}_{\mathbf{X}(T)} \odot \widetilde{\mathbf{z}}_{\mathbf{X}(T)}) \cdot \mathbf{1}_k^\top - \mathbf{1}_n(\widetilde{\mathbf{z}}_{\mathbf{X}(T)}^\top \odot \widetilde{\mathbf{z}}_{\mathbf{X}(T)}^\top) \cdot \mathbf{Y}^\top.$$

Let $\nabla_\mathbf{X} f(\mathbf{X})$ denote the exact gradient. Then we get that

$$
\begin{aligned}
\left\| \widetilde{\nabla}_\mathbf{X} f(\mathbf{X}) - \nabla_\mathbf{X} f(\mathbf{X}) \right\|_F &\leq \frac{CW}{2n}\Big[ \left\| 2 \cdot (\widetilde{\mathbf{z}}_{\mathbf{X}(T)} \cdot \widetilde{\mathbf{z}}_{\mathbf{X}(T)}^\top - \mathbf{z}_\mathbf{X} \cdot \mathbf{z}_\mathbf{X}^\top) \cdot \mathbf{Y}^\top \right\|_F \\
&\quad + \left\| (\widetilde{\mathbf{z}}_{\mathbf{X}(T)} \odot \widetilde{\mathbf{z}}_{\mathbf{X}(T)} - \mathbf{z}_\mathbf{X} \odot \mathbf{z}_\mathbf{X}) \cdot \mathbf{1}_k^\top \right\|_F \\
&\quad + \left\| \mathbf{1}_n(\widetilde{\mathbf{z}}_{\mathbf{X}(T)}^\top \odot \widetilde{\mathbf{z}}_{\mathbf{X}(T)}^\top - \mathbf{z}_\mathbf{X}^\top \odot \mathbf{z}_\mathbf{X}^\top) \cdot \mathbf{Y}^\top \right\|_F \Big] \\
&\leq \frac{CW}{2n}\Big[ 2\|\mathbf{Y}\|_F \cdot \left\| \widetilde{\mathbf{z}}_{\mathbf{X}(T)} \cdot \widetilde{\mathbf{z}}_{\mathbf{X}(T)}^\top - \mathbf{z}_\mathbf{X} \cdot \mathbf{z}_\mathbf{X}^\top \right\|_2 \\
&\quad + \sqrt{k} \cdot \left\| \widetilde{\mathbf{z}}_{\mathbf{X}(T)} \odot \widetilde{\mathbf{z}}_{\mathbf{X}(T)} - \mathbf{z}_\mathbf{X} \odot \mathbf{z}_\mathbf{X} \right\|_2 \\
&\quad + \sqrt{n}\|\mathbf{Y}\|_F \cdot \left\| \widetilde{\mathbf{z}}_{\mathbf{X}(T)}^\top \odot \widetilde{\mathbf{z}}_{\mathbf{X}(T)}^\top - \mathbf{z}_\mathbf{X}^\top \odot \mathbf{z}_\mathbf{X}^\top \right\|_2 \Big].
\end{aligned}
$$

Now let $\mathbf{e}$ be the error vector such that $\widetilde{\mathbf{z}}_\mathbf{X} = \mathbf{z}_\mathbf{X} + \mathbf{e}$ and recall that by Proposition 3 we have that $\|\widetilde{\mathbf{z}}_\mathbf{X} - \mathbf{z}_\mathbf{X}\|_2 = \|\mathbf{e}\|_2 \leq \epsilon'$. Then

$$
\begin{aligned}
\left\| \widetilde{\mathbf{z}}_{\mathbf{X}(T)} \cdot \widetilde{\mathbf{z}}_{\mathbf{X}(T)}^\top - \mathbf{z}_\mathbf{X} \cdot \mathbf{z}_\mathbf{X}^\top \right\|_2 &= \left\| (\mathbf{z}_\mathbf{X} + \mathbf{e}) \cdot (\mathbf{z}_\mathbf{X} + \mathbf{e})^\top - \mathbf{z}_\mathbf{X} \cdot \mathbf{z}_\mathbf{X}^\top \right\|_2 \\
&\leq \left\| \mathbf{e} \cdot \mathbf{e}^\top \right\|_2 + 2\left\| \mathbf{z}_\mathbf{X} \cdot \mathbf{e}^\top \right\|_2 \\
&\leq \|\mathbf{e}\|_2^2 + 2\|\mathbf{z}_\mathbf{X}\|_2 \cdot \|\mathbf{e}\|_2 \\
&\leq \|\mathbf{e}\|_2^2 + 2\sqrt{n} \cdot \|\mathbf{e}\|_2 \\
&\leq \frac{3}{8} \cdot \frac{\epsilon \cdot n}{(1+CW) \cdot (1+\|\mathbf{Y}\|_F)}.
\end{aligned}
$$

Next, using Lemma 8 we get that

$$
\begin{aligned}
\left\| \widetilde{\mathbf{z}}_{\mathbf{X}(T)} \odot \widetilde{\mathbf{z}}_{\mathbf{X}(T)} - \mathbf{z}_\mathbf{X} \odot \mathbf{z}_\mathbf{X} \right\|_2 &\leq 2\left\| \widetilde{\mathbf{z}}_{\mathbf{X}(T)} - \mathbf{z}_\mathbf{X} \right\|_2 \\
&\leq 2\|\mathbf{e}\| \\
&\leq \frac{1}{4} \cdot \frac{\min\{\epsilon, \sqrt{\epsilon}\} \cdot \sqrt{n}}{(1+CW) \cdot (1+\|\mathbf{Y}\|_F)}.
\end{aligned}
$$

Now continuing our inequalities from above and using that $k \leq n$, we obtain that

$$
\begin{aligned}
\left\| \widetilde{\nabla}_\mathbf{X} f(\mathbf{X}) - \nabla_\mathbf{X} f(\mathbf{X}) \right\|_F &\leq \frac{CW}{2n}\Big[ 2\|\mathbf{Y}\|_F \cdot \left\| \widetilde{\mathbf{z}}_{\mathbf{X}(T)} \cdot \widetilde{\mathbf{z}}_{\mathbf{X}(T)}^\top - \mathbf{z}_\mathbf{X} \cdot \mathbf{z}_\mathbf{X}^\top \right\|_2 \\
&\quad + \sqrt{k} \cdot \left\| \widetilde{\mathbf{z}}_{\mathbf{X}(T)} \odot \widetilde{\mathbf{z}}_{\mathbf{X}(T)} - \mathbf{z}_\mathbf{X} \odot \mathbf{z}_\mathbf{X} \right\|_2 \\
&\quad + \sqrt{n}\|\mathbf{Y}\|_F \cdot \left\| \widetilde{\mathbf{z}}_{\mathbf{X}(T)}^\top \odot \widetilde{\mathbf{z}}_{\mathbf{X}(T)}^\top - \mathbf{z}_\mathbf{X}^\top \odot \mathbf{z}_\mathbf{X}^\top \right\|_2 \Big] \\
&\leq \frac{CW}{2n}\left( \frac{6}{8} \cdot \frac{\epsilon \cdot n}{1+CW} + \frac{1}{4} \cdot \frac{\min\{\epsilon, \sqrt{\epsilon}\} \cdot n}{(1+CW)(1+\|\mathbf{Y}\|_F)} + \frac{1}{4} \cdot \frac{\min\{\epsilon, \sqrt{\epsilon}\} \cdot n}{1+CW} \right) \\
&\leq \frac{5}{8}\epsilon.
\end{aligned}
$$

## C.7 Proof of Theorem 6

*C.7.1 Recap of d'Aspremont's Algorithm.* We start by formally stating the result by D'Aspremont [12] for gradient descent with approximate gradient.

Suppose we wish to solve the convex optimization problem $\min_{x \in Q} f(x)$, where $f$ is a convex function mapping to $\mathbb{R}$ and $Q$ is a convex set of feasible solutions. D'Aspremont [12] gave an algorithm which approximately solves this problem using gradient descent, where the gradient contains some amount of noise. This algorithm is based on a method by Nesterov [31].

We state the pseudocode in Algorithm 4, where we let $\widetilde{T}_Q = \arg\min_{y \in Q}\left\{ \langle \widetilde{\nabla}_\mathbf{X} f(x), y - x \rangle + \frac{L}{2}\|y - x\|^2 \right\}$ and $d(x)$ is a prox-function for the set $Q$, i.e., $d$ is continuous and strongly convex with parameter $\kappa$. We state the guarantees for the algorithm in the lemma below, where $\|\cdot\|^*$ is the dual norm of $\|\cdot\|$.

---

**Algorithm 4:** Gradient-descent algorithm with noisy gradient

---

1  $x_0 \leftarrow \arg\min_{x \in Q} d(x)$

2  **for** $k = 1, \ldots, T$ **do**

3      Compute the approximate gradient $\widetilde{\nabla}_{\mathbf{X}} f(x_k)$

4      $y_k \leftarrow \widetilde{T}(x_k)$

5      $z_k \leftarrow \arg\min_{x \in Q} \{ \frac{L}{\kappa} d(x) + \sum_{i=0}^{k} \alpha_i [f(x_i) + \langle \widetilde{\nabla}_{\mathbf{X}} f(x_i), x - x_i \rangle] \}$

6      $A_k \leftarrow \sum_{i=0}^{k} \alpha_i$

7      $\tau_k \leftarrow \alpha_{k+1} / A_{k+1}$

8      $x_{k+1} \leftarrow \tau_k z_k + (1 - \tau_k) y_k$

---

**Lemma 16** (d'Aspremont [12]).  *Let $L, \kappa, \epsilon > 0$. Suppose the following conditions hold:*

*(1) $\|\nabla_{\mathbf{X}} f(x) - \nabla_{\mathbf{X}} f(y)\|^* \leq L \cdot \|x - y\|$ for all $x, y \in Q$,*

*(2) $\left| \langle \widetilde{\nabla}_{\mathbf{X}} f(x) - \nabla_{\mathbf{X}} f(x), y - z \rangle \right| \leq \epsilon$ for all $x, y, z \in Q$,*

*(3) $x_0 = \arg\min_{x \in Q} d(x)$ and $d(x_0) = 0$,*

*(4) $d(x) \geq \frac{\kappa}{2} \|x - x_0\|^2$,*

*(5) $(\alpha_k)_k$ is a sequence such that $0 < \alpha_0 \leq 1$ and $\alpha_k^2 \leq A_k$ for all $k \geq 0$.*

*Then Algorithm 4 satisfies $f(y_k) - f(x^*) \leq \frac{L d(x^*)}{A_k \kappa} + 3\epsilon$ for all $k \leq T$, where $x^* = \arg\min_{x \in Q} f(x)$ is the minimizer of the optimization problem.*

*C.7.2  Proof of Theorem 6.* Recall that we let $Q \subseteq \mathbb{R}^{n \times k}$ denote the convex subset of feasible solutions for Problem 2. We prove the following useful lemma, which shows how certain functions can be optimized over our set of constraints $Q$.

We start by proving a lemma that will be useful later when implementing d'Aspremont's algorithm.

**Lemma 17.**  *Let $\beta > 0$ and let $\mathbf{B}, \mathbf{X} \in \mathbb{R}^{n \times k}$. Then the minimizer*

$$\mathbf{X}^* = \arg\min_{\mathbf{U} \in Q} \left\{ \frac{\beta}{2} \|\mathbf{U} - \mathbf{X}\|_F^2 + \langle \mathbf{B}, \mathbf{U} \rangle_F \right\} \in \mathbb{R}^{n \times k}$$

*satisfies*

$$\mathbf{X}_i^* = \mathrm{Proj}_Q \left( \mathbf{X}_i - \frac{1}{\beta} \mathbf{B}_i - \mu_i^* \mathbf{1} \right),$$

*for all $i$. Furthermore, $\mathbf{X}^*$ can be computed in time $O(nk)$.*

Proof.  We have that

$$\mathbf{X}^* = \arg\min_{\mathbf{U} \in Q} \left\{ \frac{\beta}{2} \|\mathbf{U} - \mathbf{X}\|_F^2 + \langle \mathbf{B}, \mathbf{U} \rangle_F \right\}$$

$$= \arg\min_{\mathbf{U} \in \mathbb{R}^{n \times k}} \left\{ \delta_Q(\mathbf{U}) + \frac{\beta}{2} \|\mathbf{U} - \mathbf{X}\|_F^2 + \langle \mathbf{B}, \mathbf{U} \rangle_F \right\}$$

$$= \arg\min_{\mathbf{U} \in \mathbb{R}^{n \times k}} \left\{ \delta_Q(\mathbf{U}) + \frac{\beta}{2} \left( \langle \mathbf{U}, \mathbf{U} \rangle_F - 2\langle \mathbf{U}, \mathbf{X} \rangle_F + \langle \mathbf{X}, \mathbf{X} \rangle_F \right) + \langle \mathbf{B}, \mathbf{U} \rangle_F \right\}$$

$$= \arg\min_{\mathbf{U} \in \mathbb{R}^{n \times k}} \left\{ \delta_Q(\mathbf{U}) + \frac{\beta}{2} \left( \langle \mathbf{U}, \mathbf{U} \rangle_F - 2\langle \mathbf{U}, \mathbf{X} - \frac{1}{\beta} \mathbf{B} \rangle_F + \langle \mathbf{X}, \mathbf{X} \rangle_F \right) \right\}$$

$$= \arg\min_{\mathbf{U} \in \mathbb{R}^{n \times k}} \left\{ \delta_Q(\mathbf{U}) + \frac{\beta}{2} \left( \langle \mathbf{U}, \mathbf{U} \rangle_F - 2\langle \mathbf{U}, \mathbf{X} - \frac{1}{\beta} \mathbf{B} \rangle_F + \langle \mathbf{X}, \mathbf{X} \rangle_F - 2\langle \mathbf{X}, \frac{1}{\beta} \mathbf{B} \rangle_F + \langle \frac{1}{\beta} \mathbf{B}, \frac{1}{\beta} \mathbf{B} \rangle_F \right) \right\}$$

$$= \arg\min_{\mathbf{U} \in \mathbb{R}^{n \times k}} \left\{ \delta_Q(\mathbf{U}) + \frac{\beta}{2} \left( \langle \mathbf{U}, \mathbf{U} \rangle_F - 2\langle \mathbf{U}, \mathbf{X} - \frac{1}{\beta} \mathbf{B} \rangle_F + \langle \mathbf{X} - \frac{1}{\beta} \mathbf{B}, \mathbf{X} - \frac{1}{\beta} \mathbf{B} \rangle_F \right) \right\}$$

$$= \arg\min_{\mathbf{U} \in \mathbb{R}^{n \times k}} \left\{ \delta_Q(\mathbf{U}) + \frac{\beta}{2} \left\| \mathbf{U} - \left( \mathbf{X} - \frac{1}{\beta} \mathbf{B} \right) \right\|_F \right\}$$

$$= \arg\min_{\mathbf{U} \in \mathbb{R}^{n \times k}} \left\{ \delta_Q(\mathbf{U}) + \frac{1}{2} \left\| \mathbf{U} - \left( \mathbf{X} - \frac{1}{\beta} \mathbf{B} \right) \right\|_F \right\}$$

$$= \mathrm{prox}_{\delta_Q} \left( \mathbf{X} - \frac{1}{\beta} \mathbf{B} \right),$$

where in the penultimate step we use that $\delta_Q(\mathbf{U})$ only takes the values $0$ and $\infty$ and therefore we did not have to rescale $\delta_Q$.

Next, let us consider the projection on our set of $Q$, which is given by all $\mathbf{X}'$ such that $\left\|\mathbf{X}'_i\right\|_{1,1} = 1$ for all $i$ and $\mathbf{X}^{(L)} \leq \mathbf{X}' \leq \mathbf{X}^{(U)}$. As discussed before, the first constraint is equivalent to the hyperplane constraint $\langle \mathbf{X}'_i, \mathbf{1} \rangle = 1$ for all $i$, since $\mathbf{X}'_i \in [0, 1]^n$ and the second constraint can be rewritten as a sequence of box constraints $\mathbf{X}^{(L)}_i \leq \mathbf{X}'_i \leq \mathbf{X}^{(U)}_i$ for all $i$.

From the previous paragraph we get that it suffices if we project on the feasible set for each row of $\mathbf{X}^*$ individually. Since in the definition of $\mathrm{prox}_{\delta_Q}\left(\mathbf{X} - \frac{1}{\beta}\mathbf{B}\right)$ we considered the Frobenius norm, we can find the minimizer for each row individually. More concretely, the rows $\mathbf{X}_i$ of $\mathbf{X}^*$ are given by $\mathrm{prox}_{\delta_{Q_i}}(\mathbf{X}_i - \frac{1}{\beta}\mathbf{B}_i)$, where by $Q_i$ we denote the set of constraints $\left\|\mathbf{X}'_i\right\|_{1,1} = 1$ and $\mathbf{X}^{(L)}_i \leq \mathbf{X}'_i \leq \mathbf{X}^{(U)}_i$.

We note that this is the same as computing the orthogonal projection of $\mathbf{X}_i - \frac{1}{\beta}\mathbf{B}_i$ onto $Q_i$. This orthogonal projection can be computed by the algorithm of Kiwiel [21] in time $O(k)$. Since we have to run this procedure for each of the $n$ rows of $\mathbf{X}^*$, we can compute $\mathbf{X}^*$ in time $O(nk)$. □

To prove the theorem, we need to argue that we can apply Lemma 16 to GDPM and we also have to analyze the running time.

First, we note that GDPM is an implementation of Algorithm 4 with the following parameters.

We set $\kappa = 1$ and $d(\mathbf{X}') = \frac{1}{2}\left\|\mathbf{X}' - \mathbf{X}^{(0)}\right\|_F^2$; note that this trivially implies $d(\mathbf{X}') \geq \frac{1}{2}\kappa\left\|\mathbf{X}' - \mathbf{X}^{(0)}\right\|_F^2$ and also $d(\mathbf{X}^{(0)}) = 0$. Furthermore, we set $\alpha_T = \frac{T+1}{2}$, which satisfies the conditions of Lemma 16 as pointed out in [12]. It remains to show that $\mathbf{Z}^{(T)}$ and $\widetilde{T}(\mathbf{X}^{(T)})$ are implemented in accordance with Lemma 16. To this end, observe that

$$\mathbf{Z}^{(T)} = \underset{\mathbf{U} \in Q}{\arg\min}\left\{\frac{L}{\kappa}d(\mathbf{U}) + \sum_{t=0}^{T}\alpha_t\left[f(\mathbf{X}^{(t)}) + \langle\widetilde{\nabla}_{\mathbf{X}}f(\mathbf{X}^{(t)}), \mathbf{U} - \mathbf{X}^{(t)}\rangle_F\right]\right\}$$

$$= \underset{\mathbf{U} \in Q}{\arg\min}\left\{L\left\|\mathbf{U} - \mathbf{X}^{(0)}\right\|_F^2 + \langle\sum_{t=0}^{T}\alpha_t\widetilde{\nabla}_{\mathbf{X}}f(\mathbf{X}^{(t)}), \mathbf{U}\rangle_F\right\}.$$

Hence, we can compute $\mathbf{Z}^{(T)}$ using Lemma 17 with parameters $\mathbf{X} = \mathbf{X}^{(0)}$, $\mathbf{B} = \sum_{t=0}^{T}\alpha_t\widetilde{\nabla}_{\mathbf{X}}f(\mathbf{X}^{(t)})$, and $\beta = 2L$. Additionally, for the projection on our set of constraints we obtain that

$$\widetilde{T}_Q(\mathbf{X}^{(T)}) = \underset{\mathbf{U} \in Q}{\arg\min}\left\{\langle\widetilde{\nabla}_{\mathbf{X}}f(\mathbf{X}^{(T)}), \mathbf{U} - \mathbf{X}^{(T)}\rangle_F + \frac{L}{2}\left\|\mathbf{U} - \mathbf{X}^{(T)}\right\|_F^2\right\}$$

$$= \underset{\mathbf{U} \in Q}{\arg\min}\left\{\langle\widetilde{\nabla}_{\mathbf{X}}f(\mathbf{X}^{(T)}), \mathbf{U}\rangle_F + \frac{L}{2}\left\|\mathbf{U} - \mathbf{X}^{(T)}\right\|_F^2\right\}.$$

Hence, we can compute $\widetilde{T}_Q(\mathbf{X}^{(T)})$ using Lemma 17 with parameters $\mathbf{X} = \mathbf{X}^{(T)}$, $\mathbf{B} = \widetilde{\nabla}_{\mathbf{X}}f(\mathbf{X}^{(T)})$, and $\beta = L$.

Observe that in GDPM, $\mathbf{V}^{(T)}$ corresponds to $\widetilde{T}_Q(\mathbf{X}^{(T)})$ and $\mathbf{W}^{(T)}$ corresponds to $\mathbf{Z}^{(T)}$.

This implies that we can we can use Lemma 16 if we can also show that the gradient of our objective function is Lipschitz and that the error for our gradient is small.

First, using that we are working with the Frobenius norm which is self-dual, we can apply Proposition 5 to obtain the gradient is $L$-smooth for $L = \frac{8CW}{\sqrt{n}} \cdot \|\mathbf{s}\|_2 \cdot \|\mathbf{Y}\|_2^2$.

Second, we obtain our approximate gradient result as follows. Observe that our feasible space only contains row-stochastic matrices with entries in $[0, 1]$. Thus, we get that for all $\mathbf{X}_2, \mathbf{X}_3 \in Q$ we have that $\|\mathbf{X}_2 - \mathbf{X}_3\|_F^2 \leq 2n$, since in each row the difference can be at most 2. Thus, if we compute $\widetilde{\nabla}_{\mathbf{X}}f(\mathbf{X}_1)$ using Lemma 5 with $\epsilon' = \frac{\epsilon}{\sqrt{2n}}$ and using the Cauchy–Schwarz inequality we get that

$$\left|\langle\widetilde{\nabla}_{\mathbf{X}}f(\mathbf{X}_1) - \nabla_{\mathbf{X}}f(\mathbf{X}_2), \mathbf{X}_2 - \mathbf{X}_2\rangle_F\right|$$

$$\leq \left\|\widetilde{\nabla}_{\mathbf{X}}f(\mathbf{X}_1) - \nabla_{\mathbf{X}}f(\mathbf{X}_2)\right\|_F \cdot \|\mathbf{X}_2 - \mathbf{X}_2\|_F$$

$$\leq \epsilon' \cdot \sqrt{2n}$$

$$\leq \epsilon$$

for all $\mathbf{X}_1, \mathbf{X}_2, \mathbf{X}_3 \in Q$.

As pointed out in [12], if we compute the gradient with precision $\epsilon/6$ then we obtain a solution with additive error at most $\epsilon$ after $O\left(\frac{Ld(\mathbf{X}^*)}{\epsilon}\right)$ iterations of the algorithm, where $\mathbf{X}^*$ is the optimal solution. Since $\mathbf{X}^*$ and $\mathbf{X}^{(0)}$ are row-stochastic matrices with entries in $[0, 1]$, we get that $d(\mathbf{X}^*) \leq n$. Now using our previous bound on $L$ and $\|\mathbf{s}\|_2 \leq \sqrt{n}$, as well as $\|\mathbf{Y}\|_2^2 \leq \|\mathbf{Y}\|_F^2 \leq k$ since $\mathbf{Y}$ is row stochastic with $k$ rows, we get that the number of iterations is bounded by $O\left(\sqrt{\frac{CWkn}{\epsilon}}\right)$.

Next, observe that in each iteration we spend expected time $\widetilde{O}((mk + nk^2 + k^3)\log(W/\epsilon))$ to compute the approximate gradient by Proposition 5. Computing the matrices $\mathbf{V}^{(T)}$ and $\mathbf{W}^{(T)}$ takes time $O(nk \log k)$ by Lemma 17. Thus, the total time of each iteration is bounded by $\widetilde{O}((mk + nk^2 + k^3)\log(W/\epsilon))$.

Combing these results we obtain a total expected running time of $\widetilde{O}\left(\sqrt{\frac{CWkn}{\epsilon}}(mk + nk^2 + k^3)\log(W/\epsilon)\right)$.

