# OpenReview forum: "Modeling the Impact of Timeline Algorithms on Opinion Dynamics Using Low-rank Updates"
_ACM.org/TheWebConf/2024/Conference — TheWebConf24 Oral_

### Official Review · Reviewer_fhCx · 2023-10-26

**Novelty:** 6
**Technical Quality:** 7

**Review:**

### Summary

The authors consider the problem of reducing polarization and disagreement in a variant of the Friedkin-Johnsen (FJ) model that captures the effect of timeline recommendation algorithms on user exposure to different topics. In their FJ variant, the adjacency matrix of the graph is augmented with a weighted matrix capturing user influence via a low-rank decomposition through topics. This provides a natural way of modeling the influence of a platform's recommendation algorithm, which can increase the fraction of content a user sees on a particular topic. The authors propose an optimization problem to minimize the disagreement-polarization index of the graph with a constrained change to the user-topic exposure matrix. While the optimization problem is convex, and therefore solvable in polynomial time, naive black-box optimization is still prohibitively expensive. To address this, the authors provide an efficient algorithm to approximate the expressed opinions in their modified FJ model. Then, they develop an approximate gradient descent algorithm to efficiently solve the optimization problem. These algorithms are supported by approximation guarantees. The authors evaluate their algorithm on several real-world graphs, including a novel Twitter dataset which includes reasonable ground-truth opinions. Their approach is orders of magnitude faster than black-box optimization, which exceeds resource constraints in some cases. Moreover, it produces better solutions than two greedy baselines.


### Overall impression
Thank you to the authors for an interesting submission! I think this is a great paper, with a nice mix of theoretical guarantees and experiments. The approach to capturing timeline recommendation in the FJ model is much more natural than some of the direct edge addition approaches in prior literature, and the speed and performance of the proposed algorithm is very impressive. The new Twitter dataset is also a nice bonus. However, I think the paper would benefit from a little bit of polishing, particularly in the structure of Section 6.

### Strengths
1. The problem addressed is important and the new variant of the Friedkin-Johnsen model provides a useful new approach for tying together the theory of opinion dynamics and the practice of timeline recommendation.
2. The algorithms are supported by theoretical guarantees and perform well in practice, being much faster than naive black-box optimization and giving better results than heuristic baselines.
3. The paper is very well-written and generally quite clear.
4. The technical quality seems very good. (But, given the length of the appendix, I cannot attest to the correctness of the proofs).


### Weaknesses
1. The results section is structured awkwardly, but I think this is easy to fix (see comments below for suggestions).



### Comments
1. The matrix $M$ is used in the statement of proposition 3 is only defined in the proof sketch. It would be clearer to define $M$ alongside $U$ and $V$.

2. I'm used to smaller learning rate corresponding to smaller steps, whereas the "learning rate" in section 6 is the inverse (larger = smaller steps). My suggestion would be to redefine $L$ to what is currently $L^{-1}$, so you can call $L$ the learning rate in a more usual sense.

3. Section 6 felt out of order. I think it would be clearest to focus on results in order of importance: (1) GDPM gives good solutions, (2) it does so quickly, and (3) here's how. This would suggest putting "Comparison with greedy baselines" and "performance of the optimization algorithms" first, followed by "understanding the behavior of GDPM".

4. Along the lines of the previous comment, I don't think "Impact of learning rate" should be the first subsection in the evaluation, as this isn't really critical. This could be moved to the appendix without losing much. In fact, I think Figure 2 should be in the appendix in favor of Figure 7 that shows a more important result: that GDPM is very fast (or some other figure demonstrating fast runtime, maybe one that also includes BL-1 and BL-2).

5. I don't think Figure 1 benefits from the quadratic fits--I would suggest removing them. I also think Figure 1 should come after the figures that demonstrate good performance and runtime, as Figure 1 is about investigating how GDPM achieves its good performance (which we don't need until we've seen the good performance).

6. I found myself forgetting what $\theta$ was by the time I got to Figure 3. I would remind the reader what $\theta$ represents in the text of "comparison with greedy baselines" and/or the Figure 3 caption (a reminder about $C$ would also be useful).

7. It seems like TwitterLarge and TwitterSmall could be added to Table 2, as could the results of the black-box solver Convex.jl (if it ran out of memory or times out, then this can be noted and is even more evidence for the benefit of GDPM; or, if it achieves the same improvement as GDPM, then this attests to the correctness of your algorithm).

8. GDPM, through Algorithm 1, relies on the Solve() subroutine, which as far as I can tell isn't described in the paper, and is only mentioned to be an algorithm of Koutis, Miller and Peng in the appendix. Since Solve() is used in the proof sketch of Proposition 3, I suggest adding a citation in the main text and saying explicitly that you use as a subroutine their algorithm, which you call Solve(), and referring the reader to their paper for details.

9. In general, when referring to a figure or table in the appendix, I would suggest saying explicitly that they are located in the appendix (for instance "see Tables 3 and 2" on p.7, line 750).

**Questions:**

1. Is there any intuition for the bound on $||V M^{-1} U||_2$ in Proposition 3, or for what this matrix captures?

**Reviewer Confidence:**

3: The reviewer is confident but not certain that the evaluation is correct

**Scope:**

4: The work is relevant to the Web and to the track, and is of broad interest to the community

---

### Official Review · Reviewer_inex · 2023-10-30

**Novelty:** 5
**Technical Quality:** 6

**Review:**

The authors point out critical problems of existing studies on polarization and disagreement in online social networks, with a particular focus on the impact of timeline algorithms. As a solution, they propose an augmented FJ model that combines a fixed underlying graph with timeline algorithm-derived aggregate information. This model is optimized by a novel algorithm that guarantees bounded running time.

The proposed method is also empirically evaluated on 27 real-world datasets, including newly collected ones. Extensive experiments including important ablation studies are conducted to confirm the model's effectiveness and verify its design components. The codes are submitted and will be released to the public, and future directions proposed by the authors sound reasonable and interesting.

The paper is constructive and well-organized. However, I have some questions for the authors, which I would like to clarify.

**Questions:**

- The authors fixed C=0.1 in most of their experiments. How is the running time affected by different C values?
- The authors compared their method with two greedy baselines. It would be valuable to explore other baseline methods for a more comprehensive evaluation.
- The method in this paper is specifically rooted in the FJ model. However, it would be worthwhile to explore (or at least discuss) potential extensions of this approach to other popular algorithms.
- Have the authors considered other metrics such as internal conflict, controversy, or disagreement-controversy? Please refer to the papers below.


Quantifying and minimizing risk of conflict in social networks (KDD 2018)\
Measuring and moderating opinion polarization in social networks (Data Mining and Knowledge Discovery 2017)\
Minimizing Polarization and Disagreement in Social Networks (WWW 2018)

**Reviewer Confidence:**

2: The reviewer is willing to defend the evaluation, but it is likely that the reviewer did not understand parts of the paper

**Scope:**

4: The work is relevant to the Web and to the track, and is of broad interest to the community

---

### Official Review · Reviewer_Hcuw · 2023-11-23

**Novelty:** 5
**Technical Quality:** 4

**Review:**

In this paper, the authors augment the Friedkin-Johnson model by considering aggregated timeline algorithms to address polarization. My review focuses on the following points:

Positive Points:

1) The topic is socially relevant and important, especially in the context of responsible AI.
2) The paper proposes a gradient-based algorithm as a theoretical contribution.
3)  I recognize the theoretical results in the appendix as an important contribution. However I note that some of them are only straightforward derivations of basic linear algebra manipulations. Moreover, they are not standardly written,  defining each variable again (see, for example, C.6.) which makes harder to read.

Negative Points:

1) Regarding the organization of the paper, it needs significant improvements. While the text is easy to read, the organization of some sections is suboptimal. For instance, contributions are extensively discussed in one section, covering topics dealt with in other parts of the paper. The abstract is excessively large. The experimental section must have a better organization in flow of the ideas.

2) Some of the motivations presented by the authors lack substantiation in related works. For instance, stating that timeline algorithms are based on a user's local neighborhood without considering only local information is vague. There is an extensive related literature on personalization that deals with polarization, and these are neglected. An extended section analyzing the problem from the perspective of personalization would enhance the paper.

3) The proposed baselines seem to be very weak.  Could you better motivate the choose of the baselines?

4) Although I recognized the gradient-based algorithm as a positive aspect for optimizing the problem, I feel that the comparison with black box solvers seems overstated. Note that, while I agree that the authors' method should be more efficient over time, the conditions of comparison regarding memory are not clear.

5) Minor: There is an issue with the numeration of the paper. Lemma 1, Problem 2, Proposition 3—all of them should be numbered independently.

**Questions:**

Could you please provide more detailed explanations for the motivation behind your model, particularly concerning local-information-based timeline algorithms?

Can you clarify the conditions for comparison with optimizers? You cite Convex.jl as a black box, but it relies on popular solvers. Are they really black boxes? Could you elaborate on why, in terms of memory efficiency, your method is still more efficient than such solvers based on that discussion?

**Reviewer Confidence:**

2: The reviewer is willing to defend the evaluation, but it is likely that the reviewer did not understand parts of the paper

**Scope:**

4: The work is relevant to the Web and to the track, and is of broad interest to the community

---

### Official Review · Reviewer_ZLbT · 2023-11-23

**Novelty:** 6
**Technical Quality:** 4

**Review:**

The authors address the potential societal issues caused by existing timeline algorithms and the technical challenges of integrating different levels, namely network-level and user-level. In response to these challenges, the authors extend the existing FJ model to aggregate information from different levels. This allows for the mitigation of polarization by modifying user timelines. In this process, the GDPM is introduced, and the authors provide approximate bounds on its time complexity and solution. To foster the development of the research community, the authors have made the dataset openly accessible.

While the paper is not well-written, its content aligns with the aim and scope of WWW. Specific strengths and weaknesses are outlined as follows.

**Strengths**

+ S1: Significant research question.
+ S2: Contribution to the research community through data open access.
+ S3: Thorough theoretical validation.

The authors address the potential impact of existing timeline algorithms on polarization and disagreement in societies within social networks, which is a significant and pressing issue in contemporary society. This imbues the study with strong real-world relevance. Furthermore, the authors have made their data openly accessible, greatly expanding the existing dataset's scale. The increase in the number of nodes by one to two orders of magnitude provides a more comprehensive foundation for research in this field, making it a notable contribution. Additionally, the authors provide thorough theoretical proofs to ensure that their proposed algorithm can achieve near-linear time complexity, along with bounds on the approximation range. Such theoretical guarantees are relatively uncommon in related research.

**Weaknesses**

- W1: Lengthy and unclear abstract.
- W2: Poor paper writing.
- W3: Lack of clear overall framework.

The most significant issue in this paper is the excessively long and intricate abstract. The abstract fails to indicate the authors' work and contributions, making it challenging for readers to gauge the insights the paper offers. Similarly, the organization of the main body of the paper is problematic, with the section from lines 123 to 183 excessively elaborating on the contributions, leading to a lack of clarity. Authors should avoid overly detailed descriptions of their methods in this section. Additionally, an overarching framework diagram could help readers better understand the authors' intentions.

**Questions:**

1. The authors conducted experiments with a substantial amount of datasets, suggesting that their proposed method applies to a broad range of data distribution patterns. What mechanism enables this adaptability? Are there potential data scenarios in real-world applications where this method might face challenges in adaptation?
2. The authors mention the integration of network-level and user-level, but how is this integration specifically manifested? Are there similar mechanisms in other existing methods?

**Reviewer Confidence:**

3: The reviewer is confident but not certain that the evaluation is correct

**Scope:**

4: The work is relevant to the Web and to the track, and is of broad interest to the community

---

### Official Review · Reviewer_Dxqo · 2023-11-23

**Novelty:** 4
**Technical Quality:** 6

**Review:**

This paper introduces a method to modify the recommendations of the timeline algorithm, with the objective of reducing polarization and disagreement by allowing subtle adjustments to users' attention towards specific topics. To this end, the paper first presents a model that can quantify the impact of timeline algorithms on polarization and disagreements, and then presents a theoretical analysis of the approximation and time complexity bounds of the proposed algorithm. This paper supports future research by publishing its code and collected datasets.


Strong points.
- S1: This paper provides its code and collected datasets, surpassing the size of previously available datasets with ground-truth opinions. The accessibility of this code and dataset will prove valuable for future research.
- S2: This paper introduces a novel model based on the Friedkin-Johnsen (FJ) model, incorporating aggregate information from the timeline algorithm. This new model quantifies the influence of timeline algorithms on polarization and disagreements.
- S3: This paper introduces a gradient descent-based algorithm aimed at minimizing polarization and disagreement through minor adjustments to users' attention towards specific topics. Additionally, this paper provides theoretical approximations for the algorithm's bound and time complexity.

Weak points.

- W1: The studied problem may be appealing in theory, but it is not clear if there is any real social network that will use the proposed method to reduce polarization and disagreements.  Evidence to support the practical utility of the problem setting (i.e., Problem 2) would be great. In addition, I am wondering how well the adopted index I(G) can measure polarization in real-world applications.
- W2: This paper conducts experiences on two different parameters, C and \theta, by separately changing one while keeping the other constant. It would be a plus if this paper conducted experiments where both C and  \theta are simultaneously changed.
- W3: It would be better if the paper could explain why the first baseline fails to minimize polarization and disagreement.

**Questions:**

Please see the weak points.

**Reviewer Confidence:**

4: The reviewer is certain that the evaluation is correct and very familiar with the relevant literature

**Scope:**

3: The work is somewhat relevant to the Web and to the track, and is of narrow interest to a sub-community

---

### Decision · Program_Chairs · 2024-01-22

**Decision:**

Accept (Oral)

**Comment:**

The paper introduces a compelling variant of the Friedkin-Johnsen (FJ) model to depict the influence of timeline recommendation algorithms on user exposure to various topics. It proposes an optimization problem aiming to minimize the disagreement-polarization index and presents efficient algorithms with theoretical guarantees and strong practical performance.

 Overall, the paper offers a novel approach to an intriguing problem, supported by a balanced mix of theoretical and experimental results. However, the writing quality leaves something to be desired.